# Genomic epidemiology of putative hypervirulent *Klebsiella pneumoniae* species complex in Dutch patients, January–December 2022

Karuna E. W. Vendrik,[1] Gijs Teunis,[1] Fardau Anema,[1] Fabian Landman,[1] Angela de Haan,[1] Jeroen Bos,[1] Sandra Witteveen,[1] Annelot F. Schoffelen,[1] Sabine C. de Greeff,[1] Ed J. Kuijper,[1,2] Antoni P. A. Hendrickx,[1] Daan W. Notermans,[1] on behalf of the hvKp study group

**ABSTRACT** Hypervirulent *Klebsiella pneumoniae* species complex (hvKp) can cause invasive infections with spontaneous abscesses, also in previously healthy individuals. In contrast to Asia, hvKp is considered rare in Europe but has received more attention in the last few years, especially carbapenemase-producing strains. The aim of this prospective survey was to determine the occurrence and clinical, epidemiological, and genomic characteristics of *K. pneumoniae* species complex (KpSC) infections leading to spontaneous abscesses in Dutch patients. All Dutch medical microbiology laboratories ($n = 51$) were requested to submit KpSC isolates from 2022, that were suspected to be hypervirulent based on clinical criteria, with a spontaneous abscess as the most important criterium. Short-read sequencing (also combined with long-read sequencing for hybrid assemblies) was performed to analyze virulence factors and antimicrobial resistance genes and genetic relatedness by whole-genome multilocus sequence typing (wgMLST). In total, 33 KpSC isolates from 33 patients were submitted of whom 64% had a liver abscess and 64% had bacteremia. Among 31 patients with comorbidity information, 48% had no comorbidity. Isolates were susceptible to commonly used antibiotics. Six (18%) isolates did not have the salmochelin, yersiniabactin, aerobactin, colibactin, *rmpADC,* or *rmpA2* gene (clusters). Thirty-six percent of isolates had a maximum Kleborate virulence score. Thirty percent were ST23. WgMLST of the isolates showed low genetic relatedness compared to each other and to 720 international hypervirulent and/or ST23 KpSC isolates from NCBI. In conclusion, this study suggests that hvKp strains do occur but are relatively uncommon in Dutch patients and differ from international strains. No carbapenemase-producers were found among study isolates. When existing microbiological/molecular definitions would be used, several spontaneous abscesses could not be explained.

**IMPORTANCE** Hypervirulent *Klebsiella pneumoniae* species complex (hvKp) can lead to severe infections with abscesses in previously healthy individuals. HvKp is considered rare in Europe but has received more attention recently. A complicating factor is the absence of a clear microbiological/molecular definition of hvKp. The aim of this survey was to determine occurrence and characteristics of *K. pneumoniae* species complex (KpSC) infections leading to spontaneous abscesses, suggestive of hvKp, in Dutch patients. Dutch medical microbiology laboratories were requested to submit KpSC isolates cultured in 2022 from patients with spontaneous abscesses. This study suggests that hvKp is relatively uncommon in Dutch patients with only 33 collected isolates. The isolates were susceptible to commonly used antibiotics. Genetic characteristics were very diverse. We found low genetic relatedness compared to each other and to international hvKp isolates. When existing microbiological/molecular definitions of hvKp would be used, several spontaneous abscesses from this study could not be explained.

**Peer Reviewers** Likhona Dingiswayo, University of Free State, Bloemfontein, Free State, South Africa; Ulises Garza-Ramos, Instituto Nacional de Salud Publica, Cuernavaca, Morelos, Mexico

Address correspondence to Karuna E. W. Vendrik, karuna.vendrik@rivm.nl.

The authors declare no conflict of interest.

**KEYWORDS**   *Klebsiella pneumoniae*, ST23, hypervirulent, hvkp

*K*lebsiella pneumoniae species complex (KpSC) (1) can cause severe infections with spontaneous abscesses, most commonly observed in previously healthy individuals from the community and is then considered hypervirulent *K. pneumoniae* species complex (hvKp) (2). HvKp infections of nearly all body sites have been observed. HvKp can spread through the body in a metastatic manner. Cryptogenic abscesses have been reported at almost all body sites, with the liver being the most frequently observed location (3, 4). Besides abscesses, the most common manifestations are bacteremia, septic thromboembolism/-phlebitis, pneumonia/pleural space infections, endophthalmitis, and infections of the urinary tract, heart, central nervous system, musculoskeletal system, and soft tissue (2, 4). HvKp infections are often more severe than similar infections due to other bacteria and are mostly associated with high morbidity and mortality rates (2, 3). There is no consensus on the clinical characteristics of a hvKp infection.

There is also no consensus on the microbiological/molecular characteristics of hvKp that can distinguish hvKp from a classical *K. pneumoniae* (5, 6). A hypermucoviscous phenotype, defined by a positive string test (7), is in some literature used to define hvKp. However, hypermucoviscosity has no optimal sensitivity and specificity for detecting hvKp and a string test is, therefore, not reliable in defining hvKp (8, 9). Furthermore, certain sequence types (e.g., ST23, ST65, ST66, and ST86) and capsule types (e.g., K1 and K2) are often observed in hvKp (6), but these can also occur in non-hvKp strains and genes conferring hypervirulence are also observed in other STs and capsule types (4). *K. pneumoniae* isolates with other STs can acquire a so called hvKp virulence plasmid and become hvKp (10). A number of virulence genes present on large virulence plasmids (e.g., pK2044 and pLVPK) have been shown to perform well in differentiating hvKp from non-hvKp strains (4, 11). However, it is unclear whether the presence of these virulence plasmids alone is sufficient to confer hypervirulence (6). In addition, hypervirulence-associated virulence genes can also be present within integrated chromosomal elements (1). The Kleborate virulence score, a score based on the presence of one or more key loci associated with increasing virulence (yersiniabactin < colibactin < aerobactin), is also frequently used (12). However, there is evidence that other virulence genes not included in this score may also play a role (4, 11). It has been suggested that the murine infection model is the gold-standard experimental approach for distinguishing hypervirulent from classical strains (6, 11, 13, 14). However, this is rarely possible in routine diagnostic/surveillance settings.

The clinical syndrome of hvKp infection was first recognized in 1986 in Taiwan, where it was defined as pyogenic liver abscess associated with septic endophthalmitis (15). Since then, the number of cases with *K. pneumoniae* liver abscess in the Asian Pacific Rim has been increasing. *K. pneumoniae* liver abscess was always part of the hvKp definition, but more clinical pictures were observed that were deemed characteristic for hvKp infections, such as other septic metastatic lesions and being community-acquired. Before the 1980s, *Escherichia coli* was the most common cause of pyogenic liver abscesses in the Asian Pacific Rim, but this has gradually shifted toward *K. pneumoniae* (mostly with the capsular K1 serotype). In some East Asian regions, such as Taiwan and Korea, hvKp is now endemic (4). Besides Asia, there are reports from other continents, such as North America and Europe, indicating hvKp is also increasingly being observed in these continents (but various different definitions are being used) (4, 16, 17). In Ireland, a large outbreak of a ST23-K1 hvKp strain, the globally dominant hvKp lineage, carrying carbapenemase genes with transmission throughout a network of healthcare facilities was reported over the period of 2019–2023 (16). The European Centre for Disease prevention and Control reported in February 2024 multiple patients with infections due to ST23-K1 hvKp from numerous different European countries. Unfortunately, the number of convergent hvKp infections seems to be increasing, but again different definitions are being used (4, 16). Convergent hypervirulent *Klebsiella pneumoniae* strains are strains that combine both

hypervirulence and multidrug resistance—two characteristics that historically existed in separate lineages.

The prevalence of hvKp infections in the Netherlands is unknown. The main objective of this study was to determine occurrence and clinical, epidemiological, and genomic characteristics of KpSC infections leading to spontaneous abscesses in Dutch patients.

## RESULTS

### Occurrence

In a country with nearly 17.6 million inhabitants on 1 January 2022 (18), the National Institute for Public Health and the Environment (RIVM) had received 54 KpSC isolates from 50 patients that were suspected of having an infection with a hypervirulent strain in 2022. However, 21 isolates were excluded from the study because of duplicates ($n = 4$) or they did not meet the clinical criteria determined for this study because of the finding of other *Klebsiella* species than KpSC ($n = 2$) or because the presence of an abscess was not confirmed or unclear ($n = 15$). This means that 33 KpSC isolates from 33 patients with spontaneous abscesses were included in the analysis, from 21 laboratories with 1–4 isolates per laboratory.

The 33 isolates were initially identified as 29 *K. pneumoniae* and 4 *K. variicola* isolates using MALDI-TOF. Confirmation by next-generation sequencing (NGS; Kleborate) showed that one *K. pneumoniae* isolate actually was *K. quasipneumoniae subsp. quasipneumoniae*, which is not present in the MALDI-TOF database (Bruker).

### Epidemiological and clinical patient characteristics

Characteristics of the 33 patients are shown in Table 1. The total *N* represents the number of patients with available data. The majority (25/33; 76%) of the patients were male. More than half of the patients (21/33; 64%) had a bacteremia. One patient had a bacteremia, liver abscess, urinary tract infection, pneumonia, and meningitis due to *K. pneumoniae*. The majority of the abscesses in patients were located in the liver (21/33; 64%), followed by the lungs/pleural cavity (5/33; 15%). Multiple abscess locations were observed in at least five patients (for two additional patients, it was unclear whether they had one or more abscess locations). Almost half of the patients (15/31; 48%) had no comorbidities. The most common comorbidity was malignancy (6/31; 19%). Besides the Netherlands (11/32; 34%), the most common birthplace was Asia (6/32; 19%; including the one patient of which we know that she had recently traveled abroad [to Korea <12 months ago]). Travels outside Europe was unknown for 22 patients (69%) with one person with missing data and country of birth was unknown for 13 patients (41%) with one person with missing data.

### Hypermucoviscosity and antimicrobial resistance

Among the 33 KpSC isolates, 19 (58%) isolates were string test positive (Table 2; Table S3). The isolates were susceptible to almost all antibiotics for which information was available (Table S4). Among tested isolates, only two isolates had resistance to ciprofloxacin, and one isolate had resistance to trimethoprim/sulfamethoxazole. Carbapenem-resistance, carbapenemase production, or carbapenemase genes ($bla_{NDM}$, $bla_{KPC}$, $bla_{VIM}$, $bla_{IMP}$, or $bla_{OXA-48-like}$) were not found in the isolates (Tables S4 and S5). We also found no extended-spectrum beta-lactamase (ESBL) genes, so we did not find the most clinically relevant convergent strains. Isolates had a median of 2 (interquartile range [IQR] 2–3) resistance genes per isolate (Table S5).

### Genomic analyses of the isolates

#### Genetic relatedness

Figure 1 shows the minimum spanning trees of the 28 *K. pneumoniae* and 4 *K. variicola* isolates based on whole-genome multilocus sequence typing (wgMLST) results with

**TABLE 1** Characteristics of 33 patients with spontaneous abscesses due to *Klebsiella pneumoniae* species complex[i]

| Characteristic | n | N total (data available) | % (for age: median age [IQR[h]]) |
|---|---|---|---|
| Median age (IQR[h]) | | 33 | 68.8 (56.6–74.4) |
| Sex (% male/female) | 25/8 | 33 | 76%/24% |
| Clinical diagnosis[a] | | | |
| Abscess[b] | 33 | 33 | 100% |
| Bacteremia | 21 | 33 | 64% |
| Pneumonia/lung or pleural abscesses | 6 | 33 | 18% |
| Meningitis | 4 | 33 | 12% |
| Urinary tract infection | 2 | 33 | 6% |
| Spondylodiscitis | 2 | 33 | 6% |
| Other[c] | 2 | 33 | 6% |
| Abscess location[a] | | | |
| Liver | 21 | 33 | 64% |
| Lungs/pleural cavity | 5 | 33 | 15% |
| Brains | 2 | 33 | 6% |
| Muscle | 4 | 33 | 12% |
| Abdominal cavity | 2 | 33 | 6% |
| Bones/cartilage/soft tissue | 2 | 33 | 6% |
| Other[d] | 4 | 33 | 12% |
| Comorbidity[a] | | | |
| Malignancy | 6 | 31 | 19% |
| Immunosuppressives | 0 | 31 | 0% |
| Diabetes | 4 | 31 | 13% |
| None | 15 | 31 | 48% |
| Other comorbidities[e] | 3 | 31 | 10% |
| Unknown | 4 | 31 | 13% |
| Travels outside Europe <12 months ago | | | |
| Korea | 1 | 32 | 3% |
| None | 9 | 32 | 28% |
| Unknown | 22 | 32 | 69% |
| Country of birth | | | |
| Netherlands | 11 | 32 | 34% |
| Asia[f] | 6 | 32 | 19% |
| Africa[g] | 2 | 32 | 6% |
| Unknown | 13 | 32 | 41% |

[a]Multiple answers could be filled in so the numbers do not add up to 100%.
[b]Including one patient with an infected liver cyst and one patient with an infected kidney cyst.
[c]Peritonitis and infected shoulder prosthesis (*n* = 1 each).
[d]Epidural, testis, sinuses, kidney (*n* = 1 each).
[e]Chronic kidney insufficiency, Parkinson's disease, hypertension (*n* = 1 each).
[f]China *n* = 2, India *n* = 1, Indonesia *n* = 1, Yemen *n* = 1, Korea *n* = 1.
[g]Burundi *n* = 1, Somalia *n* = 1.
[h]IQR: interquartile range.
[i]The total *N* represents the total number of patients for which an answer to the particular question is available (including patients for which unknown was filled in).

colors representing the different MLST sequence types. The minimum allelic distance between *K. pneumoniae* isolates was 74 and between *K. variicola* 2,546. There were no genetic clusters using an allelic distance of ≤20 as cutoff. There were 10 (30%) isolates with MLST ST23, of which 8 had Kleborate virulence score 5 out of 5 (Table 2; Table S3). These 10 isolates with ST23 and the closely related ST2044 are grouped in the same part of the minimum spanning tree, indicating more genetic relatedness compared to other isolates with other sequence types.

There were two isolates with ST86 and two with ST66, also common and in literature previously associated with hypervirulence (4, 6). The common hypervirulence-associated

**TABLE 2** String test and multilocus sequence typing results per Kleborate virulence score of *Klebsiella pneumoniae* species complex isolates from patients with spontaneous abscesses

| Result | Kleborate virulence score[a] | | | | | | Total |
|---|---|---|---|---|---|---|---|
| | 0 (N = 6) | 1 (N = 6) | 2 (N = 1) | 3 (N = 4) | 4 (N = 4) | 5 (N = 12) | |
| String test | | | | | | | |
| – | 6 | 6 | | | | 2 | 14 |
| + | | | 1 | 4 | 4 | 10 | 19 |
| MLST ST | | | | | | | |
| 13 | 1 | | | | | | 1 |
| 23 | | | 1 | 1 | | 8 | 10 |
| 25 | | | | 1 | | | 1 |
| 35 | | 1 | | | | | 1 |
| 66 | | | | 1 | | 1 | 2 |
| 86 | | | | | 2 | | 2 |
| 189 | 1 | | | | | | 1 |
| 375 | | | | | 1 | | 1 |
| 380 | | | | 1 | | | 1 |
| 581 | | 1 | | | | | 1 |
| 730 | 1 | | | | | | 1 |
| 828 | | | | | 1 | | 1 |
| 882 | | 1 | | | | | 1 |
| 1553 | | 1 | | | | | 1 |
| 1591 | | | | | | 1 | 1 |
| 2042 | | | | | | 1 | 1 |
| 2044 | | | | | | 1 | 1 |
| 2726 | | 1 | | | | | 1 |
| 4192 | | 1 | | | | | 1 |
| 5069 | 1 | | | | | | 1 |
| 6231 | 1 | | | | | | 1 |
| 8481 | 1 | | | | | | 1 |

[a]Based on Kleborate: the virulence score depends on the presence of one or more key loci associated with increasing risk (yersiniabactin < colibactin < aerobactin) and the highest score of 5 when all three are present. MLST, multilocus sequence typing; ST, sequence type.

ST65 (4, 6) was not found. The other STs were diverse (19 different STs in 19 isolates) and in literature not previously associated with hvKp or uncommon among hvKp. Eleven (33%) isolates had capsular serotype K1 and seven (21%) had K2, which are the most common capsule types found in hvKp globally (4). The 15 other isolates had 13 different capsule types.

There was no genetic clustering of the study isolates with 1,701 *K. pneumoniae*, 32 *K. variicola,* and 29 *K. quasipneumoniae* isolates from the Dutch national carbapenemase-producing Enterobacterales (CPE) surveillance (Fig. S1 to S3) and the minimum allelic distance between a study isolate and a CPE isolate was 98 (which was the only *K. pneumoniae* from the CPE surveillance with virulence score 5, see discussion) for *K. pneumoniae*, 3,027 for *K. variicola* and 3,220 for *K. quasipneumoniae*.

The Dutch study isolates were compared to international *K. pneumoniae* (n = 701), *K. variicola* (n = 10), and *K. quasipneumoniae* (n = 9) isolates that were hypervirulent and/or ST23 from the National Center for Biotechnology Information (NCBI) database, also including convergent strains, by wgMLST (Fig. 2; Fig. S4 and S5; Table S6). There were no genetic clusters comprising both Dutch isolates and international isolates and the minimum allelic distance between the Dutch isolates and international isolates was 56 for *K. pneumoniae*, 2,171 for *K. variicola,* and 3,146 for *K. quasipneumoniae*.

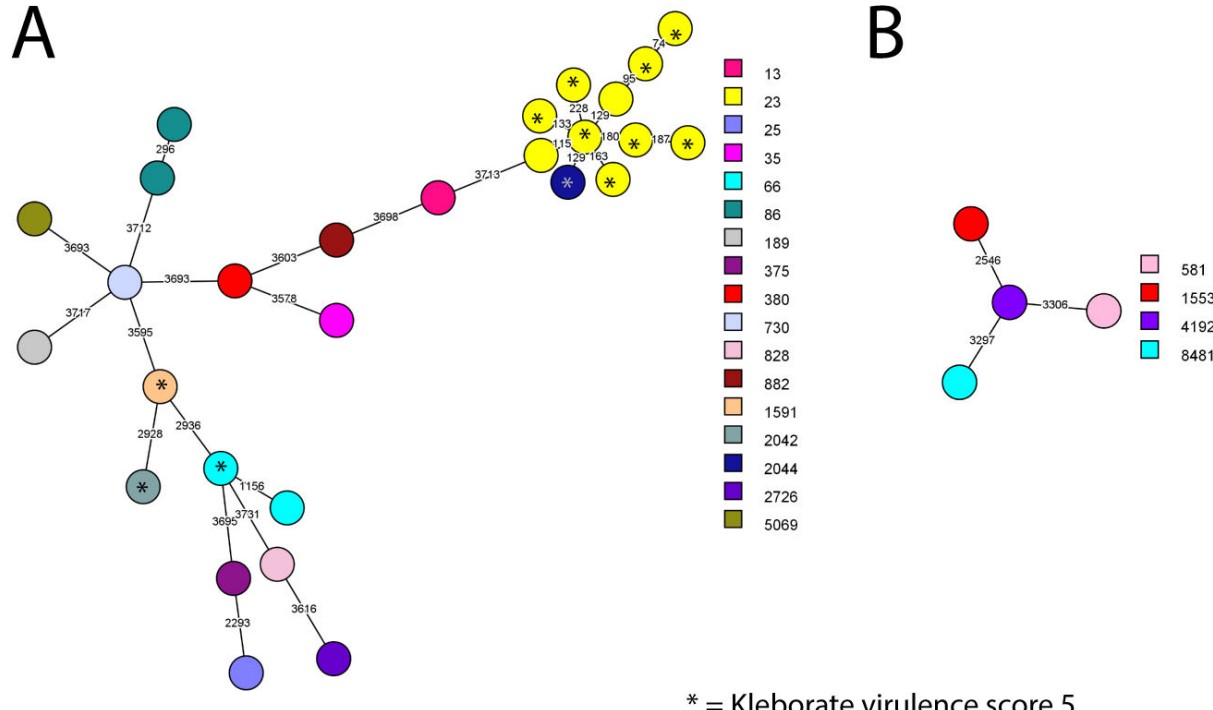

* = Kleborate virulence score 5

**FIG 1** Minimum spanning tree based on whole genome multilocus sequence typing (wgMLST) results of 28 *K. pneumoniae* (A) and 4 *K. variicola* (B) isolates from patients with spontaneous abscesses. The colors represent the MLST sequence type. A genetic cluster was defined as two or more isolates with an allelic distance of 20 or less, but no genetic clusters were found.

### Comparative genomic analysis for genes encoding putative virulence factors

Among the 33 KpSC isolates, 12 (36%) had the hypervirulence-associated maximum Kleborate virulence score of 5 (Table 2; Table S3). Among these 12 patients, seven patients (58%) had bacteremia and one (8%) had multiple abscess locations. Among 21 patients with a KpSC isolate with virulence score of 1 to 4, 14 (67%) had bacteremia and at least 4 (19%) had multiple abscess locations (two additional patients for which the number of abscess locations was unknown). Only 12 of 33 (36%) KpSC infections met the molecular hvKp definition recently proposed by Russo et al. where isolates should contain *iroB, iucA, peg-344, rmpA,* and *rmpA2* (11). These isolates had virulence scores ranging from 3 to 5. Seven isolates had both the maximum Kleborate score and met the definition of Russo et al. Ten isolates had no maximum Kleborate score and did not meet the definition of Russo et al. Differences between these two hvKp definitions were mainly due to differences in the *rmpA2* (*n* = 5) gene and the colibactin (*n* = 5) gene cluster. In total, 12 isolates did not have Kleborate virulence score 5, did not meet the definition of Russo et al., (11) were non-ST23, and had no K1 or K2 capsular serotype.

The median number of putative virulence factors was 156 (IQR 113–186) per isolate with 240 putative virulence factors being tested (Table S1; mutations in *ompK35* or *ompK36* counted as 1 if >1 mutations were present). Isolates with a virulence score 5 had a median of 197.5 (IQR 174–201) virulence factors, compared to 126 (IQR 107–158) for virulence score 1–4. This was 167 (IQR 158–199) and 109.5 (101–126) for string test-positive and -negative isolates, respectively. There were several virulence factors that were present in virtually all isolates that were not core genes based on the *K. pneumoniae sensu lato* cgMLST scheme (19) or according to Wyres et al. (1), namely genes encoding proteins involved in adhesion (*sapABC, ycfm*), *iutA* (of the aerobactin gene cluster), capsule gene expression activators (*kvrAB, rcsA), moaR,* outer membrane proteins (*kpnO/ompC^{kp}, lpp, pal),* T6SS (*tssBCDFGHJKLM, ompA*), T6SS-II (*clpV*) or T6SS-III (*impA, impF, KP1_RS15685, KPN_RS12095*) effector systems, urease (*ureBCDF*), and *ugE*. In

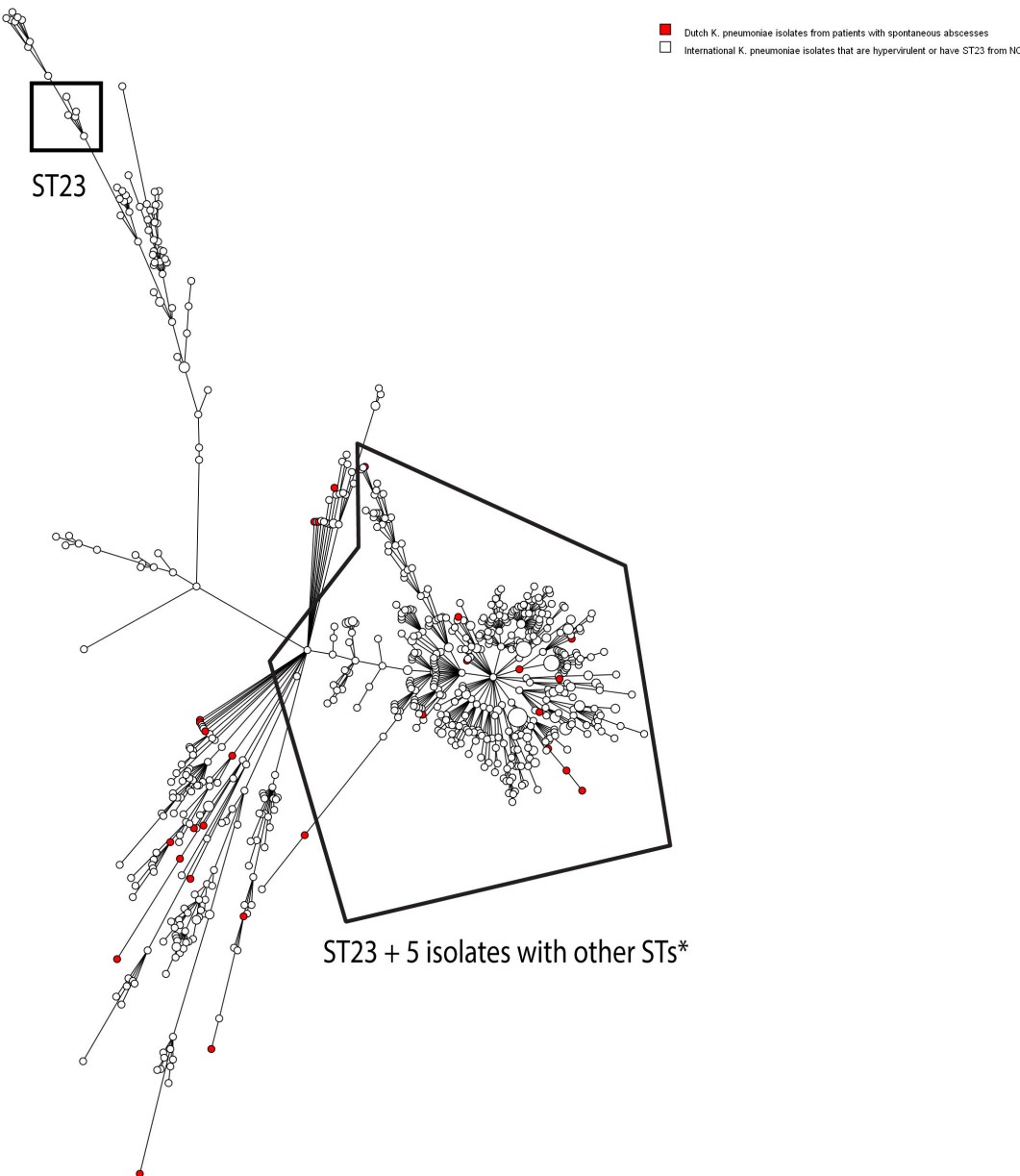

**FIG 2** Minimum spanning tree based on whole-genome multilocus sequence typing (wgMLST) results of 28 *K. pneumoniae* isolates from Dutch patients with spontaneous abscesses and 701 international hypervirulent and/or ST23 *K. pneumoniae* isolates from NCBI. The colors represent the isolate types. A genetic cluster was defined as two or more isolates with an allelic distance of 20 or less, but no genetic clusters were found including both Dutch and international isolates. Other STs*: ST1265 (*n* = 2), 1769 (*n* = 1), ST2044 (*n* = 1), non-typeable (*n* = 1).

addition, all isolates contained virulence-inhibiting factors *fnr*, (20) *fur*, (4, 20, 21), and *sugE* (22).

The colibactin gene cluster (*clbABCDEFGHILMNOPQRS*) was present in 13 (39%) isolates (of which one isolate also carried *clbJ* and two also carried *clbJ* and *clbK*). The 10 isolates without *clbJ* and *clbK* may have had a *clbJK* fusion gene caused by deletion (this was not further investigated), which may indicate loss of genotoxicity, but this is unclear (23–26). Colibactin was mainly found among ST23 isolates (*n* = 9/13). As expected, the majority of isolates had genes encoding siderophores. The salmochelin gene cluster (*iroBCDN*) was present in 23 (70%) isolates. The aerobactin gene cluster (*iucABCD, iutA*) was present in 20 (61%) isolates. The yersiniabactin gene cluster (*ybtS, ybtX, ybtQ, ybtP, ybtA, irp2, irp1, ybtU, ybtT, ybtE, fyuA*) was found in 23 (70%) isolates. The hypermucoidy

locus *rmpADC* was found in 21 (64%) isolates. Two isolates had a truncated *rmpADC* locus. The other hypermucoidy gene *rmpA2* was found in 12 (36%) isolates. Importantly, all but one *rmpA2* loci were truncated. There was one isolate with a positive string test that had no complete *rmpADC* or *rmpA2* locus (*rmpC* and *RmpD* genes were present).

There were six isolates (18%) with no yersiniabactin, aerobactin, colibactin, salmochelin, *rmpADC,* or *rmpA2* gene clusters. Virulence factors that were common in these isolates but that were not present in virtually all isolates were *fimB* (5/6), *ompK36* p.L59V (6/6), *ompK36* p.N49S (6/6), T6SS-III effector system genes (*dotU* [6/6], *impGHJ* [6/6], N559_RS31025 [5/6], *ompA* [6/6], *sciN* [6/6]), and KPNJ1_01715 (5/6). However, in total, these genes were present in >28 isolates, except for the *ompK36* mutations, which were present in 23 isolates. Since it is more likely that the five patients with multiple abscess locations had true hvKp infections, we analyzed the virulence factors in these isolates separately. These isolates contained a median of 151 (IQR 107–161) virulence genes. Virulence genes that all these isolates contained were genes that were present in virtually all study isolates. Four of the five isolates contained salmochelin, 2/5 colibactin, 2/5 aerobactin (all had *iutA*), and 4/5 yersiniabactin gene clusters and 4/5 had *rmpADC* and 0/5 *rmpA2*. Kleborate scores were 0, 1, 2, 4, and 5, and none of the isolates met the criteria of Russo et al. (11)

### *Plasmid analysis*

We found 46 plasmids among the 33 KpSC isolates from patients with spontaneous abscesses. One isolate did not harbor plasmids, one only had linear contigs, and one only had contigs <2.5 kb (Table S2). The median number of virulence genes per plasmid was 10 (IQR 0–34, range 0–35).

Three plasmids from three isolates contained resistance genes, encoding genes associated with resistance against tetracyclines, aminoglycosides, trimethoprim, and/or sulfonamides. Two of these three also contained virulence genes, encoding aerobactin, salmochelin, *rmpADC, rmpA2, peg-344/pagO, traT*, several less known genes identified as potential hvKp markers by Spadar et al. (27) and/or silver resistance.

Plasmid comparison results of plasmids from study isolates and known virulence plasmids from literature (with pLVPK, pK2044, and Kp52.145 pII being the most well-characterized virulence plasmids) (28) are shown in Table S2 and the supplemental results. Seven plasmids from seven study isolates (5 ST23 and 1 ST86 and 1 ST828) clustered with well-known virulence plasmids pLVPK(43) and pK2044(44) [also called KpVP-1(8)]. None of the Dutch plasmids clustered with the well-known Kp52.145 pII virulence plasmid. The relatedness of the Dutch and international plasmids is visualized in Fig. S6.

We found five plasmids that did not resemble any known plasmid from Enterobacterales or KpSC isolates (Table S2). Of these, two plasmids contained salmochelin and aerobactin gene clusters and *rmpADC* and *rmpA2* genes (one conjugative plasmid and one non-mobilizable *repB* plasmid). Furthermore, the first also contained the *traT* gene and aminoglycoside resistance genes and the other tellurite and silver resistance genes, and they both contained the *peg-344/pagO* gene and several less known putative virulence genes (27). One other conjugative plasmid from *K. variicola* contained only the *TraT* gene, and one *IncFIB(K)(pCAV1099-114)* non-mobilizable plasmid contained tellurite and silver resistance genes (27). The last plasmid did not contain virulence factors or resistance genes.

## DISCUSSION

This study suggests that hvKp infections do occur but are relatively uncommon in Dutch patients. In total, 33 KpSC isolates from 33 patients with spontaneous abscesses were detected and submitted during the survey in 2022. The isolates were genetically diverse and did not cluster with KpSC isolates from the Dutch CPE surveillance or with international hypervirulent and/or ST23 KpSC isolates. The well-known ST23, that is in literature associated with hypervirulence, was found in 30% of the isolates. Thirty-six

percent had the hypervirulence-associated maximum Kleborate virulence score of 5 and 36% met the hvKp definition of Russo et al. (presence of *iroB, iucA, peg-344, rmpA,* and *rmpA2*) (11). Seven isolates met both of these hvKp criteria. Some plasmids from the study isolates were related to well-known virulence plasmids, but we also identified four novel putative virulence plasmids that did not resemble a known virulence plasmid and two were conjugative.

This study was initiated after the reports of outbreaks with ST23-K1 CP hvKp in Ireland since march 2019 (29). The low genetic relatedness between the KpSC isolates from Dutch patients with spontaneous abscesses suggests there were no hvKp outbreaks or transmissions in the Netherlands in 2022. Furthermore, we did not find a carbapene-mase-producing isolate among the isolates submitted for this study. However, one CP *K. pneumoniae* with $bla_{NDM}$ and Kleborate virulence score 5 that was ST23 was found in the national CPE surveillance (30) in 2022, from a patient who had bacteremia and prostate abscesses. It did not form a genetic cluster with the study isolates. ECDC (16) and WHO (5) reports suggest that (CP) hvKp is emerging in Europe and globally. In 2018–2023, numerous different European countries reported patients with infections due to ST23-K1 hvKp isolates, the globally dominant hvKp lineage. A substantial part contained carbapenemase-genes (16). The current study suggests that hvKp infections were uncommon in Dutch patients in 2022. Two recent studies suggested that hvKp also occurs in Germany but is uncommon (31, 32), with 21% detected ST23 among hvKp (32). ST23 was the most common ST among the Dutch isolates from this study (30%), but we did not find Germany's second most common ST, ST395. In contrast to the Netherlands, carbapenemase production was more frequently found among hvKp in Germany, but comparison of the Dutch and German isolates was difficult/not possible as the inclusion criteria were different.

To date, there is no consensus regarding the definition of hvKp (5, 6). When differ-ent definitions are used, statements on and comparisons of incidence/prevalence/occur-rence of hvKp are difficult. In this study, we used a combination of clinical and molecular criteria, but other studies often based the hvKp definition on the presence of a combi-nation of certain virulence factors, a Kleborate virulence score of 5, detection of ST23, murine infection models, and/or capsule type K1 or K2. A positive string test was also previously used, but there is strong evidence that this is not reliable in defining hvKp (8, 9). There is no consensus regarding the clinical characteristics of a hvKp infection. Furthermore, the clinical picture is not always clear. The presence of a (cryptogenic) abscess is not always known, because the information is not collected (e.g., for routine laboratory surveillance without clinical information), it is not (yet) assessed or unclear or abscesses are still small or have not occurred yet. Furthermore, patients with hvKp may show other clinical pictures than an abscess, that we used as inclusion criterium, such as pneumonia (4). Another important point is that the clinical picture depends on a combination of the virulence of the KpSC and patient characteristics. Non-hvKp may also cause abscesses in patients with certain underlying diseases. Murine infection models for identifying hvKp are rarely possible in diagnostic/surveillance settings. With a proper molecular/microbiological definition, hvKp can be more closely and effectively monitored, e.g., this may lead to earlier and more diagnostics for abscesses. The results of this study suggest that existing microbiological/molecular definitions of hvKp may not cover the total problem of hvKp. Kleborate virulence score, siderophores, colibac-tin, hypermucoidy loci, sequence type, and capsule type all appear to show a weak correlation with clinical symptoms of patients with spontaneous abscesses although numbers in our study were too low for statistical analysis. A high Kleborate virulence score was often found in isolates from patients that did not have the most severe clinical picture, whereas a low virulence score was also common in patients with a severe clinical picture. The Kleborate virulence score varied among patients with multiple abscess locations that are more likely to have true hvKp infections (score of 0, 1, 2, 4, 5). Only 12 of the 33 KpSC isolates from patients with spontaneous abscesses in our study met the molecular hvKp definition proposed by Russo et al., (11) of which seven also had

Kleborate virulence score 5. Among patients with multiple abscess locations, none met the definition of Russo et al. Importantly, it is doubtful whether isolates with a low Kleborate virulence score and that do not meet the definition proposed by Russo et al. are true hvKp.

There were several virulence factors that were present in virtually all isolates and that were no core genes. Furthermore, six isolates with no siderophore, colibactin, *rmpADC,* or *rmpA2* gene clusters (including one *K. quasipneumoniae* and one *K. variicola*) often contained *fimB,* ompK36 p.L59V, ompK36 p.N49S, several T6SS-III effector system genes, and KPNJ1_01715. It is possible that one of these virulence genes or gene clusters may contribute to hypervirulence, but it is more likely that a combination of virulence factors is needed. However, this study lacked a comparator non-hvKp group, did not include *in vivo* or *in vitro* experiments using isolates with and without these genes, and did not apply a clear definition of hvKp. Therefore, no firm conclusions can be drawn.

The inclusion criteria of this study included the presence of spontaneous abscesses because these are typical for hvKp infections. However, severe community-acquired pneumonia (CAP) due to hvKp without the presence of abscesses has also been described and would have been missed (33, 34). We included six patients in this study with a pneumonia or abscess in the lungs or pleural cavity (18%), of which one without abscesses in the lungs/pleural cavity (with abscesses in another location) and five with abscesses in the lungs/pleural cavity. HvKp may cause severe CAP associated with higher rates of respiratory failure, bilateral lobar involvement, septic shock, multiorgan failure, and death, compared to other causes of CAP (4, 33, 34). Furthermore, hospital-acquired ventilator-associated pneumonia, empyema, and septic pulmonary embolism due to hvKp have also been reported (4).

For the literature search, we included all virulence factors that have been considered virulence factors or pathogenicity factors in previous studies. However, evidence for the association with hypervirulence is scarce or contradicting for some virulence factors, e.g. for *peg-344* (1).

Interestingly, we found four patients with spontaneous abscesses due to *K. variicola* and one with *K. quasipneumoniae*, that contained several virulence factors (range 92–120), including one *K. variicola* isolate that contained among others the salmochelin and yersiniabactin gene clusters and *rmpADC*. One *K. variicola* and one *K. quasipneumoniae* did not contain yersiniabactin, salmochelin, aerobactin, or colibactin gene clusters or *rmpADC* or *rmpA2*, so other isolate or patient factors may have caused the clinical picture. There is little information on how often hypervirulence is observed in these species (35).

An important limitation is that, for a large proportion of the patients, foreign travel and country of birth were unknown. Based on this study, we cannot make firm conclusions on whether hvKp is endemic in the Netherlands. Information on the severity of the disease was limited and information on whether the infection was community- or hospital-acquired was not available in this study. Furthermore, convergent hvKp strains in which the pathogenicity is so strongly affected by the addition of antimicrobial resistance genes that abscesses are no longer present are not included. Finally, we did not include a control group or perform *in vitro* or *in vivo* studies to confirm hypervirulence or analyze associations of certain virulence factors with hypervirulence.

In conclusion, this study suggests that hvKp strains do occur but are relatively uncommon in Dutch patients. KpSC isolates from patients with spontaneous abscesses were genetically diverse, with a large variety in sequence types and virulence genes. The most clinically relevant convergent hvKp strains, including CPE and ESBL-producers, were not found. When Kleborate virulence score, the proposed definition of Russo et al. (11), ST23, or capsular serotype was used to define hvKp molecularly/microbiologically, many spontaneous abscesses could not be explained, i.e., 12 when combined and 15–23 when used separately. Furthermore, 18% of the isolates missed genes encoding siderophores, colibactin, or hypermucoidy loci. The large diversity in hvKp makes a molecular/microbiological definition for hvKp difficult, and clinical information may also be crucial.

## MATERIALS AND METHODS

We performed a 1 year prospective survey in 2022. All 51 Dutch medical microbiology laboratories (MMLs) were requested to participate in this project.

### Bacterial isolates

Dutch MMLs were asked to submit KpSC isolates (see supplemental material) (1), which were suspected to be hypervirulent based on clinical criteria, to the National Institute for Public Health and the Environment. The clinical criteria were (i) KpSC cultured as only pathogen from a spontaneously occurring abscess from one or more body sites (liver, kidney, eye, spleen, brain, lung, spinal/epidural, muscle) in the absence of previous bile duct problems, perforation, or surgery or (ii) KpSC cultured from blood, in a patient with an abscess in one of the before mentioned body sites, not accessible for drainage, and without the above mentioned other explanations for an abscess. Meeting the clinical criteria was the inclusion criterium. Exclusion criteria were duplicates of the same person, non-KpSC species at confirmation and an unclear or not-confirmed presence of an abscess. The submission of isolates was voluntary.

### Epidemiological and clinical patient characteristics

A web-based questionnaire was used to collect clinical and epidemiological patient data.

### String test

Isolates were subjected to the string test, to test for hypermucoviscosity, as previously described (7).

### Antimicrobial susceptibility testing

Species identification was performed by MALDI-TOF (Microflex LT System; Bruker, Leiderdorp, Netherlands) and confirmed by NGS (Kleborate). A meropenem Etest (BioMérieux Inc., Marcy L'Étoile, France) was performed to test for carbapenem-resistance. Phenotypical carbapenemase production was assessed by using the carbapenem inactivation method (36). Furthermore, phenotypical susceptibility testing results (susceptible/susceptible with increased exposure/resistant) for nine selected antibiotics (see Table S4), generated by the submitting laboratories in routine diagnostic procedures using EUCAST recommendations and cutoffs (37), were collected via the web-based questionnaire.

### Genomic analyses

Details of the genomic analysis are reported in the supplemental material. All isolates included in this study were subjected to paired-end NGS, performed on the Illumina NextSeq550 platform (Illumina, USA). NGS results were imported into BioNumerics version 7.6.3 (Applied Maths, Sint-Martens-Latem, Belgium) for analyses.

For the assessment of virulence factors, sequences of virulence genes in *K. pneumoniae* of BIGSdb-Pasteur (38, 39) and the virulence factor database (40, 41) were used. In addition, an extensive literature search on virulence genes in *K. pneumoniae* was performed in February 2023. All isolates were screened for these putative virulence genes using BLAST v2.13.0 (42). For certain virulence factors, AMRfinder (43) and Pointfinder (44) were used (see supplemental material). The examined 240 putative virulence genes/factors and three putative virulence-inhibiting genes (*FNR* [20], *fur* [4, 20, 21], and *sugE* [22]) can be found in Table S1. Kleborate v2.3.2 was used to assess the species, virulence scores, capsular serotypes (K-antigen), and LPS serotypes (O-antigen), and predictions on virulence factor truncation were used (12). Two important molecular definitions of hvKp are Kleborate virulence score (12) and hvKp criteria recently proposed by Russo et al. (11). The Kleborate virulence score describes the presence of key

hypervirulence loci and ranges from 0 to 5: 0 = no yersiniabactin, colibactin, or aerobactin; 1 = yersiniabactin only; 2 = yersiniabactin and colibactin (or colibactin only); 3 = aerobactin without yersiniabactin or colibactin; 4 = aerobactin with yersiniabactin (no colibactin); 5 = yersiniabactin, colibactin, and aerobactin (12). Isolates were also checked for the hvKp criteria from Russo et al. (11). Isolates were screened for antimicrobial resistance genes with the ResFinder (45) software.

For classical MLST, the existing schemes available via SeqSphere were used. For wgMLST, minimum spanning trees were contemplated using BioNumerics version 8.1 using an in-house wgMLST scheme (46). A genetic cluster was defined as two or more isolates with an allelic distance of ≤20 (47). Furthermore, the genomes of the isolates were compared to genomes of hypervirulent (also including other STs than ST23) and/or ST23 KpSC isolates from the NCBI database (Table S6). Furthermore, the isolates were compared to KpSC isolates from the CPE surveillance of the National Institute for Public Health and the Environment (30) to examine genetic relatedness with this large sequenced collection of (mostly classical) KpSC isolates from the Netherlands. These 1,762 sequenced CPE isolates from the period February 2012 until October 2025 included only four *K. pneumoniae* isolates with maximum Kleborate virulence score 5.

PlasmidFinder software (48) was used to assess the presence of plasmid replicons. Third-generation sequencing (TGS) via Nanopore long-read sequencing was performed to assess characteristics of plasmids. Hybrid assemblies (49) were performed by combining NGS and TGS data. Non-circular contigs and contigs of <2.5 kb were excluded. Plasmids were characterized via MOB-suite v3.1.8 (50, 51). Plasmids of isolates included in this study were compared to each other and to international virulence plasmids from the literature (search in August 2023; Table S2) using "chromosome comparison" in BioNumerics. In addition, a figure was contemplated visualizing the relatedness of Dutch and international plasmids using Average Nucleotide Identity (determined using pyANI) (52).

Dutch plasmids were compared to previously found plasmids in Enterobacterales isolates (or in KpSC isolates only in case of an error for all Enterobacterales) using NCBI BLAST on 23 October 2024 (42, 53).

## Data presentation and statistics

Data were presented as *n* (%) in case of categorical variables. Numerical variables were presented as mean (SD) or, in case of a skewed distribution, median (IQR). Due to the low number of isolates, no statistical tests were performed, but the data were described. STATA SE version 18.0 (StataCorp, College Station, TX, USA) was used for data analysis.

### ACKNOWLEDGMENTS

The authors would like to thank Selina van der Vliet for her support in project management.

This research received no specific grant from any funding agency in the public, commercial, or not-for-profit sectors.

The study was conceived and supervised by D.W.N. and A.P.A.H., supported by E.J.K. F.A. and D.W.N. drafted the study protocol and coordinated the study. K.E.W.V. coordinated the last part of the study, collected the data, analyzed the data, and wrote the manuscript. G.T. and F.L. performed the BLAST on virulence factors and G.T. made the plasmid comparison figure. A.D.H. and J.B. coordinated the laboratory experiments. A.D.H., S.W., and F.L. performed assemblies on the NGS and TGS data and made the sequencing data ready for analyses. A.F.S. and S.C.D.G. provided advice on the study protocol and epidemiological analysis. All authors critically reviewed the manuscript. The hvKp study group collected and sent the isolates and provided isolate and patient data.

The hvKp study group consists of the following: Rob J Rentenaar, University Medical Center Utrecht, Utrecht, the Netherlands; Patrick D. J. Sturm, Laurentius Hospital, Roermond, the Netherlands; Annemieke Bloem, Amphia Hospital, Breda, the Netherlands; James W. T. Cohen Stuart, Noord West Ziekenhuisgroep, Alkmaar, the

Netherlands; Maurits van Meer, Rijnstate Hospital, Arnhem, the Netherlands; Suzan van Mens, Maastricht University Medical Center+, Maastricht, the Netherlands; Michael P. M. van der Linden, IJsselland Hospital, Capelle aan den Ijssel, the Netherlands; Pieter Smit, Maasstad Hospital, Rotterdam, the Netherlands; David Dekker, Tergooi Hospital, Hilversum, the Netherlands; Gert Blaauw, Gelre Hospitals, Apeldoorn, the Netherlands; Vishal Hira, Groene Hart Hospital, Gouda, the Netherlands; Erik Bathoorn, University Medical Center Groningen, the Netherlands; Erik van der Vorm, Reinier de Graaf Hospital, Delft, the Netherlands; Harold F. J. Thiesbrummel, OLVG, Amsterdam, the Netherlands; Saara J. Vainio, Antonius Hospital, Utrecht, the Netherlands; Sunita Paltansing, Franciscus Gasthuis, Rotterdam, the Netherlands; Eva Kolwijck, Jeroen Bosch Hospital, 's Hertogenbosch, the Netherlands; Julia da Silva, Deventer Hospital, Deventer, the Netherlands; Karola Waar, Certe Medical Microbiology Friesland, Noordoostpolder, Leeuwarden, the Netherlands; Mireille van Westreenen, Erasmus Medical Center, Rotterdam, the Netherlands; Boulos Maraha, Albert Schweitzer Hospital, Dordrecht, the Netherlands; Jarne van Hattem, Amsterdam University Medical Center, Amsterdam, the Netherlands; Lieke L. Reubsaet, Haaglanden Medical Center, Den Haag, the Netherlands; Alewijn Ott, Certe Medical Microbiology Groningen, Drenthe, Groningen, the Netherlands; Arjan Jansz, Eurofins-PAMM, Veldhoven, the Netherlands; Man Chi Wong, Haga Hospital, Den Haag, the Netherlands.

## AUTHOR AFFILIATIONS

[1]Centre for Infectious Disease Control, National Institute for Public Health and the Environment (Rijksinstituut voor Volksgezondheid en Milieu, RIVM), Bilthoven, the Netherlands
[2]Department of Medical Microbiology, Leiden University Medical Center, Leiden, the Netherlands

## PRESENT ADDRESS

Fardau Anema, Department of Medical Microbiology, Alrijne hospital, Leiden, the Netherlands
Ed J. Kuijper, Department of Medical Microbiology, Leiden University Medical Center, Leiden, the Netherlands

## AUTHOR ORCIDs

Karuna E. W. Vendrik  http://orcid.org/0000-0002-9259-4920
Sandra Witteveen  http://orcid.org/0000-0001-9132-9028
Ed J. Kuijper  http://orcid.org/0000-0001-5726-2405
Antoni P. A. Hendrickx  http://orcid.org/0000-0003-4431-9323

## DATA AVAILABILITY

Raw NGS sequence data of all study isolates were deposited in the Sequence Read Archive, and the plasmids were deposited in GenBank of NCBI under BioProject ID PRJNA1284600 (see Tables S1 and S2 for accessions).

## ETHICS APPROVAL

KpSC isolates and patient data collected in this study were obtained as part of routine clinical care. All data were collected from the clinical electronic health records and laboratory information systems. No additional patient samples or patient data were requested from the patients specifically for this study and no actions were requested from patients. The data were pseudonymized. Investigators did not have access to personally identifiable information of the patients. Written or verbal informed consent was therefore not required. This study did not fall under the scope of the Dutch Medical Research Involving Human Subjects Act (WMO) and was, therefore, exempt from review by an Institutional Review Board.

## ADDITIONAL FILES

The following material is available online.

### Supplemental Material

**Supplemental figures (Spectrum02259-25-s0001.pdf).** Fig. S1 to S6.
**Supplemental material (Spectrum02259-25-s0002.pdf).** Supplemental text.
**Table S1 (Spectrum02259-25-s0003.xlsx).** Virulence genes.
**Table S2 (Spectrum02259-25-s0004.xlsx).** Plasmid characteristics.
**Table S3 (Spectrum02259-25-s0005.xlsx).** Isolate characteristics.
**Table S4 (Spectrum02259-25-s0006.xlsx).** Phenotypic testing results.
**Table S5 (Spectrum02259-25-s0007.xlsx).** Antimicrobial resistance genes.
**Table S6 (Spectrum02259-25-s0008.xlsx).** Accessions from international isolates and plasmids.

### Open Peer Review

**PEER REVIEW HISTORY (review-history.pdf).** An accounting of the reviewer comments and feedback.

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
