## [Reviewer comments · Microbiology Spectrum]

Microbiology Spectrum

Genomic epidemiology of putative hypervirulent *Klebsiella pneumoniae* species complex in Dutch patients, January-December 2022

Karuna Vendrik, Gijs Teunis, Fardau Anema, Fabian Landman, Angela de Haan, Jeroen Bos, Sandra Witteveen, Annelot Schoffelen, Sabine de Greeff, Ed Kuijper, Antoni Hendrickx, and Daan Notermans

Corresponding Author(s): Karuna Vendrik, Rijksinstituut voor Volksgezondheid en Milieu

Review Timeline:

Submission Date:	July 23, 2025
Editorial Decision:	August 26, 2025
Revision Received:	November 25, 2025
Editorial Decision:	November 28, 2025
Revision Received:	December 1, 2025
Accepted:	December 4, 2025

Editor: Gabriele Arcari

Reviewer(s): Disclosure of reviewer identity is with reference to reviewer comments included in decision letter(s). The following individuals involved in review of your submission have agreed to reveal their identity: Likhona Dingiswayo (Reviewer #1); Ulises Garza-Ramos (Reviewer #2)

Transaction Report:

DOI: <https://doi.org/10.1128/spectrum.02259-25>

Re: Spectrum02259-25 (**Genomic epidemiology of hypervirulent *Klebsiella pneumoniae* complex in Dutch patients, January-December 2022**)

Dear Dr. Karuna E.W. Vendrik:

Thank you for the privilege of reviewing your work. Below you will find my comments, instructions from the Spectrum editorial office, and the reviewer comments.

Revision Guidelines

Sincerely,
Gabriele Arcari
Editor
Microbiology Spectrum

Reviewer #1 (Comments for the Author):

1. The study is well-conducted, explains all the important points, and provides further guidance that hypervirulent *Klebsiella pneumoniae* (hvKp) is a minimal concern in the Netherlands.
2. Line 228 states "The 33 presumed hvKp isolates concerned 29 *K. pneumoniae* and four *K. variicola* isolates (based on 228

MALDI-TOF)". Revise the highlighted word.

3. From Lines 444-447, the study mentions that no comparison was done to deduce whether the presence/absence of these virulent associated determinants. I strongly agree and therefore a set of known non-hvKp isolates and sequencing was needed for such comparison. This would assist and give more guidance in defining hvKp. Also, a pathogenic response (in vivo and invitro models) of hvKp that had the presence of siderophore, colibactin, rmpADC or rmpA2 vs those that had no presence of these genes e.g K. quasipneumoniae and one K. variicola.

Reviewer #2 (Comments for the Author):

The study describes the genomic characterization of isolates of the presumably hypervirulent K. pneumoniae species complex. The initial classification for including the isolates in the analysis was based on clinical criteria, with their origin from an abscess as the main criterion, and later genomic criteria were used. However, although the strains caused abscesses, genetically they do not meet the criteria to be considered hypervirulent. Likewise, no experimental characterization was performed to evaluate their pathogenicity.

The study is relevant from the perspective of collecting clinical isolates causing various abscesses from patients hospitalized in various hospitals across the country. However, I believe that the question or objective of the study should be reconsidered. This is because the fact that a strain causes an abscess does not necessarily mean that it must be hypervirulent. Perhaps the strains that caused metastasis in patients should be evaluated; these would have a greater chance of being hypervirulent, for that reason alone. It is understood that the definition of hypervirulence has changed over time; however, there are basic aspects that a strain must meet to be considered hypervirulent, ranging from having hypervirulence genes (contained in a plasmid) to having caused high mortality in experimental trials in laboratory mice. Likewise, the study should include analysis of variants that have been described in the literature in recent years; for example, convergent strains.

General Questions:

The introduction is described in reverse, i.e., the general questions at the end.

Line 79: must be Klebsiella pneumoniae species complex (KpSC). Review Wyres, K. L., Lam, M. M. C. & Holt, K. E. Population genomics of Klebsiella pneumoniae. Nat. Rev. Microbiol. 18, 344-359 (2020).

Line 144. The inclusion criteria for strains should be specified in the M&M.

Line 125: The meaning of RIVM is unclear.

Lines 153-154. It is unclear which tests it refers to.

Lines 160-175. It is important to identify the different virulence factors. Several platforms and assays are mentioned; however, Kleborate is designed for the identification of virulence factors related to Hv strains. What is the difference between the other systems, or what does the inclusion of canonical virulence factors contribute to the analysis if they are not ultimately discussed or contributed to the results or analysis?

Kleborate 0-5 should describe their nature.

Line 189: Table S2, why should the rmpACD operon genes be separated?

Results:

Line 222. The exclusion criteria are unclear. Likewise, the work actually identifies 12 potential HVs overall. The rest lack the genes described in Hv strains, and Kleborate confirms this with its virulence score of 0.1.2.

MALDI identifies 29 Kp and 4 Kv, and they mention that NGS identifies 28 Kpn, 4 Kv, and 1 Kq. Specifically, what analysis was used with the genome to identify the bacterial species? ANI?

Figure 1 does not contribute much to the analysis. According to the study, the relationship between the presumed 12 Hv (score 5) and the others considered Hv could be seen. Based on the data presented, there are doubts about whether they can be considered Hv. Likewise, the use of MLST for Kpn for Kv is not reliable. There is evidence that does not contribute to your analysis (see doi: <https://doi.org/10.1101/2024.10.28.617863> or <https://www.biorxiv.org/content/10.1101/2024.10.28.617863v1>). Likewise, the Kv MLST system should be used for the assignment of Kv STs. Likewise, include genomes of Hv strains described in other regions of the world for both Kq and Kv for comparison.

When a ST is new, you should request the assignment of the new ST from the curators of the corresponding database.

Lines 303-306: Is this an analysis of classical Hv vs. strains? What is the purpose of the comparison? This is unclear.

Line 308: Where are the Kq and Kv genomes? Likewise, these species should be compared separately with their HV genomes described in other studies.

Line 320. In your results, you describe what I mentioned earlier: identifying 12 strains as presumptive Hv strains. You describe 33 strains, as clearly showing that only these 12 met the criteria for presumptive Hv strains. Likewise, these strains should at least be characterized for their pathogenicity in a murine model, phagocytosis by macrophages, and resistance to human serum. Only then can they be confirmed as HV strains.

I consider a hypervirulent K. variicola strain with a score of zero to be highly unlikely. In the case of K. quasipneumoniae strains that are described as HV, they have acquired both the virulence factors and the capsular serotype in their chromosome.

Is there a relationship between string+ strains with the complete rmpACD operon? It should be noted that there is a third phenotype that few mention: rmpACD-negative strains that are positive to the string test. Do these strains exist in this study?

Discussion.

Various criteria have been described to effectively define an Hv strain; however, there is a strong consensus that the "murine infection model is the gold-standard experimental approach for distinguishing hypervirulent from classical strains." Likewise, the use of the string test was effectively ruled out. The use of clinical data is now an interesting proposal, but it lacks clear criteria, not because of the data itself; rather, because the evolution of Hv KpSC is currently underway, and significant discrepancies between virulence factors and pathogenicity in vivo have been reported. Likewise, the convergence phenomenon is an important factor to consider, adding variability, which would be the low pathogenicity in MDR Hv-KpSc strains. Ultimately, the discussion should be adjusted according to the considerations of the review. The data presented on presumably hypervirulent *K. pneumoniae* species complex in your country appear to be typical hypervirulent strains, it shouldn't be much of a problem to characterize them in greater depth.

The aim of this study was to determine the occurrence and clinical, epidemiological, and genomic characteristics of hvKp in Dutch patients. Presumed hvKp definition was based on clinical criteria with a cryptogenic abscess in patients.

1. The study is well-conducted and effectively explains all the important points, and provides further guidance that hypervirulent *Klebsiella pneumoniae* (hvKp) is a minimal concern in the Netherlands.
2. Line 228 states “The 33 presumed hvKp isolates **concerned** 29 *K. pneumoniae* and four *K. variicola* isolates (based on 228 MALDI-TOF)”. Revise the highlighted word.
3. From Lines 444-447, the study mentions that no comparison was done to deduce whether the presence/absence of these virulent associated determinants. I strongly agree and therefore a set of known non-hvKp isolates and sequencing was needed for such comparison. This would assist and give more guidance in defining hvKp. Also, a pathogenic response (in vivo and invitro models) of hvKp that had the presence of siderophore, colibactin, *rmpADC* or *rmpA2* vs those that had no presence of these genes e.g *K. quasipneumoniae* and one *K. variicola*.

Dear Gabriele Arcari,

Thanks you for the review of our article Spectrum02259-25 entitled "Genomic epidemiology of putative hypervirulent *Klebsiella pneumoniae* species complex in Dutch patients, January-December 2022"

We have carefully discussed all suggestions provided by the reviewers and reply point by point below. When a sentence is cited from the manuscript, the underlined part of the sentence is newly added to the manuscript.

In general, we feel that the quality of the manuscript increased considerably and we hope that the manuscript is suitable for publication in its current form.

Yours sincerely,

On behalf of all co-authors of this manuscript,

Karuna Vendrik, MD PhD

Reviewer #1 (Comments for the Author):

1. The study is well-conducted, explains all the important points, and provides further guidance that hypervirulent *Klebsiella pneumoniae* (hvKp) is a minimal concern in the Netherlands.

Reply: We thank the reviewer for his positive review.

2. Line 228 states "The 33 presumed hvKp isolates concerned 29 *K. pneumoniae* and four *K. variicola* isolates (based on 228 MALDI-TOF)". Revise the highlighted word.

Reply: We do not see a highlighted word. We assume that you mean the word 'concerned' so we have now changed this in the 'Occurrence' section of the results: 'The 33 isolates were initially identified as 29 *K. pneumoniae* and four *K. variicola* isolates using MALDI-TOF.'

3. From Lines 444-447, the study mentions that no comparison was done to deduce whether the presence/absence of these virulent associated determinants. I strongly agree and therefore a set of known non-hvKp isolates and sequencing was needed for such comparison. This would assist and give more guidance in defining hvKp. Also, a pathogenic response (in vivo and invitro models) of hvKp that had the presence of siderophore, colibactin, rmpADC or rmpA2 vs those that had no presence of these genes e.g *K. quasipneumoniae* and one *K. variicola*.

Reply: We agree with the reviewer that this is a limitation. A good comparator group of non-hvKp isolates was not available and this was therefore not possible, but, as the reviewer mentions, we have already added this limitation to the discussion.

We agree that experimental characterization is needed to confirm whether the strains are or are not hypervirulent. Unfortunately, we are not able to do *in vivo* or *in vitro* experiments and

this was beyond the scope of this manuscript. Therefore we have not drawn hard conclusions on hvKp.

We have now added the limitation to the discussion that *in vivo* or *in vitro* model studies are also needed: *‘However, this study lacked a comparator non-hvKp group, did not include in vivo or in vitro experiments using isolates with and without these genes, and did not apply a clear definition of hvKp. Therefore, no firm conclusions can be drawn.’*

Furthermore, we have also added this sentence to the limitation section in the discussion: *‘Finally, we did not include a control group or perform in vitro or vivo studies to confirm hypervirulence or analyze associations of certain virulence factors with hypervirulence.’*

Reviewer #2 (Comments for the Author):

The study describes the genomic characterization of isolates of the presumably hypervirulent *K. pneumoniae* species complex. The initial classification for including the isolates in the analysis was based on clinical criteria, with their origin from an abscess as the main criterion, and later genomic criteria were used. However, although the strains caused abscesses, genetically they do not meet the criteria to be considered hypervirulent. Likewise, no experimental characterization was performed to evaluate their pathogenicity.

Reply: Thank you for your valuable feedback. We agree that experimental characterization is needed to confirm whether the strains are or are not hypervirulent. Unfortunately, we are not able to do *in vivo* or *in vitro* experiments and this was beyond the scope of this manuscript. Therefore we have not drawn hard conclusions on hvKp.

We have now removed the term ‘hvKp’ as much as possible for the isolates from this study throughout the manuscript and added the term ‘*putative*’ to the title.

We have now added this sentence to the introduction: *‘It has been suggested that the murine infection model is the gold-standard experimental approach for distinguishing hypervirulent from classical strains.(6, 11, 13, 14) However, this is rarely possible in routine diagnostic/surveillance settings’*

We added this to the discussion: *‘other studies often based the hvKp definition on the presence of a combination of certain virulence factors, a Kleborate virulence score of 5, detection of ST23, murine infection models and/or capsule type K1 or K2.’* and *‘Murine infection models for identifying hvKp are rarely possible in diagnostic/surveillance settings.’*

And this sentence to the limitation section in the discussion: *‘Finally, we did not perform in vitro or vivo studies to confirm hypervirulence or analyze associations of certain virulence factors with hypervirulence.’*

We have also added the limitation to the discussion that *in vivo* or *in vitro* studies are needed for conclusions on hypervirulence associations: *‘However, this study lacked a comparator non-hvKp group, did not include in vivo or in vitro experiments using isolates with and without these genes, and did not apply a clear definition of hvKp. Therefore, no firm conclusions can be drawn.’*

The study is relevant from the perspective of collecting clinical isolates causing various abscesses from patients hospitalized in various hospitals across the country. However, I believe that the question or objective of the study should be reconsidered. This is

because the fact that a strain causes an abscess does not necessarily mean that it must be hypervirulent. Perhaps the strains that caused metastasis in patients should be evaluated; these would have a greater chance of being hypervirulent, for that reason alone. It is understood that the definition of hypervirulence has changed over time; however, there are basic aspects that a strain must meet to be considered hypervirulent, ranging from having hypervirulence genes (contained in a plasmid) to having caused high mortality in experimental trials in laboratory mice. Likewise, the study should include analysis of variants that have been described in the literature in recent years; for example, convergent strains.

Reply: We agree that if a strain causes an abscess it does not necessarily mean that it must be hypervirulent. For this reason we had excluded isolates from non-spontaneous abscesses, such as related to recent surgery, but we agree that other patient factors may play a role in the development of an abscess. In the discussion, we have drawn no hard conclusions on hvKp and mentioned that the clinical picture depends on a combination of the virulence of the KpSC and patient characteristics.

We have now changed the objective of the study in the abstract, importance and introduction sections, e.g.: *'The main objective of this study was to determine occurrence and clinical, epidemiological and genomic characteristics of KpSC infections causing spontaneous abscesses in Dutch patients.'*

We have also changed the conclusions in the abstract, importance section and at the start and end of the discussion to match the objective, e.g.: *'When existing microbiological/molecular definitions would be used, several spontaneous abscesses could not be explained.'* and *'In total, 33 KpSC isolates from 33 patients with spontaneous abscesses were detected'* and *'When Kleborate virulence score, the proposed definition of Russo et al., (11) ST23 or capsular serotype were used to define hvKp molecularly/microbiologically, many spontaneous abscesses could not be explained.'*

We have now added this to the discussion: *'Non-hvKp may also cause abscesses in patients with certain underlying diseases'*

Furthermore, we have now removed the term 'hvKp' as much as possible for the isolates from this study throughout the manuscript and added the term '*putative*' to the title.

We only had five patients for which it was confirmed that there was more than one abscess location. We agree that for these isolates it is more likely that these are hvKp than for isolates that cause only one abscess. We have now added a section on these isolates to section 'Comparative genomic analysis for genes encoding putative virulence factors' in the results:

'Since it is more likely that the five patients with multiple abscess locations had true hvKp infections, we analyzed the virulence factors in these isolates separately. These isolates contained a median of 151 (IQR 107-161) virulence genes. Virulence genes that all these isolates contained were genes that were present in virtually all study isolates. Four of the five isolates contained salmochelin, 2/5 colibactin, 2/5 aerobactin (all had iutA) and 4/5 yersiniabactin gene clusters and 4/5 had rmpADC and 0/5 rmpA2. Kleborate scores were 0,1,2,4 and 5 and none of the isolates met the criteria of Russo et al.'

And this to the discussion:

'The Kleborate virulence score varied among patients with multiple abscess locations, that are more likely to have true hvKp infections (score of 0, 1, 2, 4, 5).'

'Among patients with multiple abscess locations, none met the definition of Russo et al..'

We agree that detection of hypervirulence genes and observed high mortality in experimental trials in laboratory mice are both needed to confirm hypervirulence (see answer to previous question above). However, this was beyond the scope of this manuscript.

The manuscript already includes analysis of recent variants, like convergent strains, but we agree that this can be expanded/emphasized:

- We already described plasmids from our isolates with both virulence factors and resistance genes (in section 'Plasmid analyses' of the results):
'Three plasmids from three isolates contained resistance genes, encoding genes associated with resistance against tetracyclines, aminoglycosides, trimethoprim and/or sulfonamides. Two of these three also contained virulence genes, encoding aerobactin, salmochelin, rmpADC, rmpA2, peg-344/pagO, traT, several less known genes identified as potential hvKp markers by Spadar et al.(40) and/or silver resistance.'
We already compared our plasmids to all plasmids from international *K. pneumoniae* species complex isolates with available sequence data via NCBI BLAST, which also includes plasmids from convergent strains.
- We did not find the clinically most relevant convergent strains having ESBL or carbapenemase genes as was already mentioned (in section 'Hypermucoviscosity and antimicrobial resistance' of the results), but we have added text to make this more clear: *'The isolates were susceptible to almost all antibiotics for which information was available (supplementary table 4). Among tested isolates, only two isolates had resistance to ciprofloxacin and one isolate had resistance to trimethoprim/sulfamethoxazole. Carbapenem-resistance, carbapenemase production or carbapenemase genes (*bla*_{NDM}, *bla*_{KPC}, *bla*_{VIM}, *bla*_{IMP}, or *bla*_{OXA-48}) were not found the isolates (supplementary table 4 and 5). We also found no extended-spectrum beta-lactamase (ESBL) genes, so we did not find the most clinically relevant convergent strains.'*
- Furthermore, we did compare our isolates with international hvKp or ST23 *K. pneumoniae* isolates (see M&M and figure 2), which also include several convergent strains like carbapenemase-producing hvKp, and we did not find a genetic cluster. We have now added this to emphasize this in the section 'Genetic relatedness' of the results:
*'The Dutch study isolates were compared to international *K. pneumoniae* (n=701), *K. variicola* (n=10) and *K. quasipneumoniae* (n=9) isolates that were hypervirulent and/or ST23 from the NCBI database, also including convergent strains, by wgMLST'*
- We have now added that this also included non-ST23 isolates in the section 'Genomic analysis' of M&M:
'Furthermore, the genomes of the isolates were compared to genomes of hypervirulent (also including other STs than ST23) and/or ST23 KpSC isolates from the National Center for Biotechnology Information (NCBI) database'
- We have now also added the ST23 hvKp isolates (including carbapenemase-producing hvKp) from the Irish outbreak, since these sequence data were not available yet at the time we searched for sequences. (Genomic and phylogenetic analysis of hypervirulent *Klebsiella pneumoniae* ST23 in Ireland | Microbiology Society).

Furthermore, we have added Illumina sequence data from some recent studies (including carbapenemase-producing hvKp):

Genomically defined hypervirulent Klebsiella pneumoniae contributed to early-onset increased mortality | Nature Communications

Deciphering the relative importance of genetic elements in hypervirulent Klebsiella pneumoniae to guide countermeasure development - ScienceDirect

Differentiation of hypervirulent and classical Klebsiella pneumoniae with acquired drug resistance | mBio

- We have now added this text to the introduction:
‘Unfortunately, the number of convergent hvKp infections seems to be increasing, but again different definitions are being used.(3, 16) Convergent hypervirulent Klebsiella pneumoniae strains are strains that combine both hypervirulence and multidrug resistance — two characteristics that historically existed in separate lineages.’
And this to section ‘Hypermucoviscosity and antimicrobial resistance’ of Results:
‘so we did not find the most clinically relevant convergent strains.’
And this to the conclusion in the discussion:
‘The most clinically relevant convergent hvKp strains, including CPE and ESBL, were not found.’

General Questions:

The introduction is described in reverse, i.e., the general questions at the end.

Reply: In our opinion, the general question should be put at the end, because the text above gives a general introduction on the topic and missing/contradicting information, which then result in the objective of the study.

Line 79: must be Klebsiella pneumoniae species complex (KpSC). Review Wyres, K. L., Lam, M. M. C. & Holt, K. E. Population genomics of Klebsiella pneumoniae. Nat. Rev. Microbiol. 18, 344-359 (2020).

Reply: Thank you for pointing this out, we have changed *Klebsiella pneumoniae* complex into *Klebsiella pneumoniae* species complex (KpSC) throughout the manuscript.

Line 144. The inclusion criteria for strains should be specified in the M&M.

Reply: We added the in- and exclusion criteria to the section ‘Bacterial isolates’ of M&M: *‘Meeting the clinical criteria was the inclusion criterium. Exclusion criteria were duplicates of the same person, non-KpSC species at confirmation and an unclear or not-confirmed presence of an abscess.’*

Line 125: The meaning of RIVM is unclear.

Reply: We replaced RIVM throughout the manuscript by *‘National Institute for Public Health and the Environment’*.

Lines 153-154. It is unclear which tests it refers to.

Reply: We tried to clarify this in the ‘Antimicrobial susceptibility testing’ section of M&M of the manuscript: *‘Furthermore, phenotypical susceptibility testing results (susceptible/susceptible with increased exposure/resistant) for nine selected antibiotics (see*

table 4), generated by the submitting laboratories in routine diagnostic procedures using EUCAST recommendations and cut-offs, were collected via the web-based questionnaire.'

Lines 160-175. It is important to identify the different virulence factors. Several platforms and assays are mentioned; however, Kleborate is designed for the identification of virulence factors related to Hv strains. What is the difference between the other systems, or what does the inclusion of canonical virulence factors contribute to the analysis if they are not ultimately discussed or contributed to the results or analysis? Kleborate 0-5 should describe their nature.

Reply: we noticed that in other articles on hvKp several different virulence factors and databases with virulence factors were analyzed. We were interested in all potential virulence factors that are described for *K. pneumoniae*, because it is not completely clear in our opinion which virulence factors contribute to hypervirulence. We examined all results to see whether we saw noteworthy associations. In our opinion, even if we did not find it relevant to discuss certain genes in the manuscript text, it could still be relevant to other researchers to add the results (it is part of the supplementary information, not of the main text and tables).

We don't know why certain virulence genes are added to the BIGSdb and not to the virulence factor database or the other way around, but our aim was to analyze all virulence factors in *K. pneumoniae*. Since these databases did not include the same virulence factors, we added both.

The nature of Kleborate score 0-5 is now further clarified in the section 'genomic analyses' in M&M: *'The Kleborate virulence score describes presence of key hypervirulence loci and ranges from 0 to 5: 0 = no yersiniabactin, colibactin or aerobactin; 1 = yersiniabactin only; 2 = yersiniabactin and colibactin (or colibactin only); 3 = aerobactin without yersiniabactin or colibactin; 4 = aerobactin with yersiniabactin (no colibactin); 5 = yersiniabactin, colibactin and aerobactin.'*

Line 189: Table S2, why should the rmpACD operon genes be separated?

Reply: we understand that these are part of the same operon, but we chose to mention the separate genes for all gene clusters and operons. We have now placed *rmpA*, *rmpC* and *rmpD* next to each other in table S1 and table S2 to make more clear these belong together.

Results:

Line 222. The exclusion criteria are unclear.

Reply: We added the in- and exclusion criteria to the section 'Bacterial isolates' of M&M: *'Meeting the clinical criteria was the inclusion criterium. Exclusion criteria were duplicates of the same person, non-KpSC species at confirmation and an unclear or not-confirmed presence of an abscess.'*

Likewise, the work actually identifies 12 potential HVs overall. The rest lack the genes described in Hv strains, and Kleborate confirms this with its virulence score of 0.1.2.

Reply: I am not sure which criteria you mean in defining those 12 hvKp, I assume you mean the criteria proposed by Russo *et al.*, which we have mentioned in the result section. There are differences between the definition proposed by Russo *et al.* and the Kleborate score. Therefore, in the results and discussion, we mention the results of different available

molecular hvKp criteria. If you think a molecular definition is missing, we would be happy to add that to the manuscript.

We have now removed the term 'hvKp' as much as possible for the isolates from this study throughout the manuscript and added the term '*putative*' to the title.

We have emphasized that the Kleborate score and the definition proposed by Russo *et al.* are important molecular definitions of hvKp, by adding the underlined words to the section 'genomic analyses' in the M&M:

'Two important molecular definitions of hvKp are Kleborate virulence score (11) and hvKp criteria recently proposed by Russo et al.(11) The Kleborate virulence score describes the presence of key hypervirulence loci' and 'Isolates were also checked for the hvKp criteria from Russo et al.'

and by adding the underlined words to the section 'Comparative genomic analysis for genes encoding putative virulence factors' of the results (and by starting this section with both of these definitions):

'Among the 33 KpSC isolates, 12 (36%) had the hypervirulence-associated maximum Kleborate virulence score of 5.' and 'Only twelve of 33 (36%) KpSC infections met the molecular hvKp definition recently proposed by Russo et al. where isolates should contain *iroB*, *iucA*, *peg-344*, *rmpA* and *rmpA2*.(11) These isolates had virulence scores ranging from 3 to 5. Seven isolates had both the maximum Kleborate score and met the definition of Russo et al. Ten isolates had no maximum Kleborate score and did not meet the definition of Russo et al. Differences between these two hvKp definitions were mainly due to differences in the *rmpA2* (n=5) gene and the colibactin (n=5) gene cluster.'

And by adding this to the discussion:

'Thirty-six percent had the hypervirulence-associated maximum Kleborate virulence score of 5 and 36% met the hvKp definition of Russo et al (*iroB*, *iucA*, *peg-344*, *rmpA* and *rmpA2*).(11) Seven isolates met both of these hvKp criteria.'

'Importantly, it is doubtful whether isolates with a low Kleborate virulence score and that do not meet the definition proposed by Russo et al. are true hvKp'

'When Kleborate virulence score, the proposed definition of Russo et al.,(10) ST23 or capsular serotype were used to define hvKp molecularly/microbiologically, many spontaneous abscesses could not be explained'

MALDI identifies 29 Kp and 4 Kv, and they mention that NGS identifies 28 Kpn, 4 Kv, and 1 Kq. Specifically, what analysis was used with the genome to identify the bacterial species? ANI?

Reply: We used the Kleborate result. We have now added that to the manuscript in the 'Antimicrobial susceptibility testing' and 'Genomic analyses' section of M&M and 'Occurrence' section of the results.

Figure 1 does not contribute much to the analysis. According to the study, the relationship between the presumed 12 Hv (score 5) and the others considered Hv could be seen. Based on the data presented, there are doubts about whether they can be considered Hv.

Reply: We respectfully disagree with the statement that figure 1 does not contribute to the analysis. In our opinion, this is a valuable figure, most importantly because *K. pneumoniae* species complex isolates from patients with spontaneous abscesses do not form a genetic cluster, so there seems to be no outbreak in the Netherlands.

We have now added an asterisk to visualize the isolates with Kleborate score 5 in figure 1.

We added this to the discussion: 'Importantly, it is doubtful whether isolates with a low Kleborate virulence score and that do not meet the definition proposed by Russo et al. are true hvKp'

Likewise, the use of MLST for Kpn for Kv is not reliable. There is evidence that does not contribute to your analysis (see doi:

<https://www.biorxiv.org/content/10.1101/2024.10.28.617863v1> or

<https://www.biorxiv.org/content/10.1101/2024.10.28.617863v1>). Likewise, the Kv MLST system should be used for the assignment of Kv STs. Likewise, include genomes of Hv strains described in other regions of the world for both Kq and Kv for comparison.

Reply: As described in the M&M, for MLST, we use schemes from SeqSphere. These include an MLST scheme from Institut Pasteur that is designed for all species within *K. pneumoniae* species complex, including *K. variicola*: <https://bigsd.bpasteur.fr/klebsiella/>. We use this scheme and this is an internationally accepted scheme. SeqSphere does not include a scheme that is specifically developed for *K. variicola* and we found only 4 *K. variicola* isolates. It is beyond the scope of this study to develop a new scheme specifically for *K. variicola*.

For wgMLST, we also did not use a *K. variicola* specific-scheme, we used our in-house *K. pneumoniae* scheme. We have now added wgMLST data of *K. variicola* and *K. quasipneumoniae* isolates from NCBI and from the CPE surveillance to the manuscript. All species from *K. pneumoniae* species complex are checked for the percentage of detected core genes of the wgMLST scheme as quality control. WgMLST for species from *K. pneumoniae* species complex is only performed for isolates that have >90% of the core genes of the scheme. All *K. variicola* and *K. quasipneumoniae* isolates from this pilot (n=4 and n=1, resp.) and from our carbapenemase-producing surveillance (n=32 and n=29, resp.) and the international *K. variicola* isolates (n=10 and n=9 resp.) have >90% (*K. variicola* even >96%) of the core genes from our wgMLST scheme. Two *K. quasipneumoniae* isolates from our surveillance (*K. pneumoniae* in MALDI-TOF) were excluded from the wgMLST analysis due to a percentage detected core genes of 89.2% and 89.9%.

For the plasmid comparison with international plasmids, we also used NCBI BLAST en already included all KpSC species in our search.

We have now added separate minimum spanning trees for *K. pneumoniae*, *K. variicola* and *K. quasipneumoniae* using our in-house wgMLST scheme for KpSC (Figure 1 and 2 and Supplementary figures 1 to 5).

To clarify this, we added this to the 'Supplementary material and results' file: For classical multi-locus sequence typing (MLST), the existing KpSC schemes available via SeqSphere were used. For whole-genome wgMLST, minimum spanning trees were contemplated using BioNumerics version 8.1 using an in-house K. pneumoniae wgMLST scheme.(14) K. variicola and K. quasipneumoniae isolates were only included in the wgMLST analysis if they had >90% detected core genes of K. pneumoniae of the wgMLST scheme (two K. quasipneumoniae isolates from the CPE surveillance were excluded from the analysis due to a percentage of detected core genes of 89.2 % and 89.9%).

When a ST is new, you should request the assignment of the new ST from the curators of the corresponding database.

Reply: we have now requested the assignment of the new ST and changed ‘ new ST’ into ‘ST8481’ throughout the manuscript, figures and supplementary tables.

Lines 303-306: Is this an analysis of classical Hv vs. strains? What is the purpose of the comparison? This is unclear.

Reply:

This is the only large collection of sequenced *K. pneumoniae* species complex isolates from the Netherlands that we have at our institute. This included 1,762 sequenced CPE isolates (1,701 *K. pneumoniae*, 32 *K. variicola* and 29 *K. quasipneumoniae*) from the period February 2012 until October 2025. For the national carbapenemase-producing Enterobacterales surveillance, we also assess Kleborate virulence score. The large majority will probably be classical *K. pneumoniae*, only 4 CPE isolates (all *K. pneumoniae*) had Kleborate virulence score 5. We added these isolates to see whether there are genetic clusters of hvKp strains with these classical (or maybe hvKp) strains. We wondered whether there were convergent hvKp strains amongst the sequenced CPE isolates. We used wgMLST to analyze clusters.

We tried to explain this in the ‘genomic analyses’ section of the M&M: ‘Furthermore, the isolates were compared to KpSC isolates from the carbapenemase-producing Enterobacterales (CPE) surveillance of the National Institute for Public Health and the Environment(31) to examine genetic relatedness with this large sequenced collection of (mostly classical) KpSC isolates from the Netherlands. These 1,762 sequenced CPE isolates from the period February 2012 until October 2025 included only four *K. pneumoniae* isolates with maximum Kleborate virulence score 5.’

Line 308: Where are the Kq and Kv genomes? Likewise, these species should be compared separately with their HV genomes described in other studies.

Reply: See previous answer

Line 320. In your results, you describe what I mentioned earlier: identifying 12 strains as presumptive Hv strains. You describe 33 strains, as clearly showing that only these 12 met the criteria for presumptive Hv strains. Likewise, these strains should at least be characterized for their pathogenicity in a murine model, phagocytosis by macrophages, and resistance to human serum. Only then can they be confirmed as HV strains.

Reply: see previous answers on the definition of Russo *et al.*, Kleborate and experimental characterization.

I consider a hypervirulent *K. variicola* strain with a score of zero to be highly unlikely. In the case of *K. quasipneumoniae* strains that are described as HV, they have acquired both the virulence factors and the capsular serotype in their chromosome.

Reply: We agree it is unlikely that a *K. variicola* is hypervirulent when it has a Kleborate virulence score of 0. This isolate did not have any of the genes of the definition of Russo *et al* (*iroB*, *iucA*, *peg-344*, *rmpA* and *rmpA2*). It has no aerobactin, salmochelin, colibactin or yersiniabactin gene clusters, or *rmpADC* or *rmpA2*, but 99 other putative virulence genes. We

have now removed the term 'hvKp' as much as possible for the isolates from this study throughout the manuscript and added the term 'putative' to the title.

The *K. quasipneumoniae* isolate also has virulence score 0 and has several virulence genes, but not aerobactin, salmochelin, yersiniabactin, colibactin gene clusters, or *rmpA* or *rmpA2*. It did not have any of the genes of the definition of Russo et al (*iroB*, *iucA*, *peg-344*, *rmpA* and *rmpA2*). We found 4 plasmids in the *K. quasipneumoniae* isolate, among which one had putative virulence factors (only tellurite and silver resistance genes and a hypothetical protein).

Is there a relationship between string+ strains with the complete rmpACD operon? It should be noted that there is a third phenotype that few mention: rmpACD-negative strains that are positive to the string test. Do these strains exist in this study?

Reply: I had previously added this to the manuscript, but I removed it because the manuscript was already so long and it is already known that there is a relation between string test results and the rmpACD operon.

Among 19 isolates with a positive string test, 18 (95%) had a complete *rmpADC* operon, compared to 3 (21%) among 14 isolates with a negative string test (two isolates with truncated *rmpADC* locus were removed). There was one isolate with a positive string test that had no complete *rmpADC* operon (*rmpC* and *RmpD* genes were present, *rmpA2* was also not present).

I have now added this sentence to the section 'Comparative genomic analysis for genes encoding putative virulence factors' of the results: 'There was one isolate with a positive string test that had no complete rmpADC or rmpA2 locus (rmpC and RmpD genes were present).'

Discussion.

Various criteria have been described to effectively define an Hv strain; however, there is a strong consensus that the "murine infection model is the gold-standard experimental approach for distinguishing hypervirulent from classical strains."

Reply: see previous answer on experimental characterization.

Likewise, the use of the string test was effectively ruled out.

Reply: I had already mentioned this in the introduction: *'However hypermucoviscosity has no optimal sensitivity and specificity for detecting hvKp and a string test is therefore not reliable in defining hvKp.(7, 8).'*

I have now also added this sentence to the discussion and removed the string test from the rest of the discussion: 'A positive string test was also previously used, but there is strong evidence that this is not reliable in defining hvKp.(7, 8).'

The use of clinical data is now an interesting proposal, but it lacks clear criteria, not because of the data itself; rather, because the evolution of Hv KpSC is currently

underway, and significant discrepancies between virulence factors and pathogenicity in vivo have been reported.

Reply: We agree that hv KpSC are changing. Virulence genes causing hypervirulence can also change. In addition, insights into virulence factors and definitions will change. Clinical data and molecular data will probably never completely match. In our opinion, this study adds valuable information to the available literature. Unfortunately, we are not able to do *in vivo* or *in vitro* experiments, but we have now added this limitation (as described above).

Likewise, the convergence phenomenon is an important factor to consider, adding variability, which would be the low pathogenicity in MDR Hv-KpSc strains.

Reply: We did not find MDR Hv-KpSC strains (no carbapenemase or ESBL genes were found). We acknowledge that antimicrobial resistance could affect the virulence of a hvKp. However, we have only included KpSC isolates from patients with abscesses, so convergent hvKp strains in which the pathogenicity is so strongly affected that abscesses are no longer present are not included.

We have now added this to the discussion:

'Furthermore, convergent hvKp strains in which the pathogenicity is so strongly affected by addition of antimicrobial resistance genes that abscesses are no longer present are not included.'

For further discussion about convergence, see our previous answer.

Ultimately, the discussion should be adjusted according to the considerations of the review.

The data presented on presumably hypervirulent *K. pneumoniae* species complex in your country appear to be typical hypervirulent strains, it shouldn't be much of a problem to characterize them in greater depth.

Reply: see previous answers.

Re: Spectrum02259-25R1 (**Genomic epidemiology of putative hypervirulent *Klebsiella pneumoniae* species complex in Dutch patients, January-December 2022**)

Dear Dr. Karuna E.W. Vendrik:

Revision Guidelines

Sincerely,
Gabriele Arcari
Editor
Microbiology Spectrum

Dear Gabriele Arcari,

Thanks you for the review of our article Spectrum02259-25 entitled "Genomic epidemiology of putative hypervirulent *Klebsiella pneumoniae* species complex in Dutch patients, January-December 2022"

We have carefully discussed all suggestions provided by the reviewers and reply point by point below. When a sentence is cited from the manuscript, the underlined part of the sentence is newly added to the manuscript.

In general, we feel that the quality of the manuscript increased considerably and we hope that the manuscript is suitable for publication in its current form.

Yours sincerely,

On behalf of all co-authors of this manuscript,

Karuna Vendrik, MD PhD

Reviewer #1 (Comments for the Author):

1. The study is well-conducted, explains all the important points, and provides further guidance that hypervirulent *Klebsiella pneumoniae* (hvKp) is a minimal concern in the Netherlands.

Reply: We thank the reviewer for his positive review.

2. Line 228 states "The 33 presumed hvKp isolates concerned 29 *K. pneumoniae* and four *K. variicola* isolates (based on 228 MALDI-TOF)". Revise the highlighted word.

Reply: We do not see a highlighted word. We assume that you mean the word 'concerned' so we have now changed this in the 'Occurrence' section of the results: 'The 33 isolates were initially identified as 29 *K. pneumoniae* and four *K. variicola* isolates using MALDI-TOF.'

3. From Lines 444-447, the study mentions that no comparison was done to deduce whether the presence/absence of these virulent associated determinants. I strongly agree and therefore a set of known non-hvKp isolates and sequencing was needed for such comparison. This would assist and give more guidance in defining hvKp. Also, a pathogenic response (in vivo and invitro models) of hvKp that had the presence of siderophore, colibactin, rmpADC or rmpA2 vs those that had no presence of these genes e.g *K. quasipneumoniae* and one *K. variicola*.

Reply: We agree with the reviewer that this is a limitation. A good comparator group of non-hvKp isolates was not available and this was therefore not possible, but, as the reviewer mentions, we have already added this limitation to the discussion.

We agree that experimental characterization is needed to confirm whether the strains are or are not hypervirulent. Unfortunately, we are not able to do *in vivo* or *in vitro* experiments and

this was beyond the scope of this manuscript. Therefore we have not drawn hard conclusions on hvKp.

We have now added the limitation to the discussion that *in vivo* or *in vitro* model studies are also needed: *‘However, this study lacked a comparator non-hvKp group, did not include in vivo or in vitro experiments using isolates with and without these genes, and did not apply a clear definition of hvKp. Therefore, no firm conclusions can be drawn.’*

Furthermore, we have also added this sentence to the limitation section in the discussion: *‘Finally, we did not include a control group or perform in vitro or vivo studies to confirm hypervirulence or analyze associations of certain virulence factors with hypervirulence.’*

Reviewer #2 (Comments for the Author):

The study describes the genomic characterization of isolates of the presumably hypervirulent *K. pneumoniae* species complex. The initial classification for including the isolates in the analysis was based on clinical criteria, with their origin from an abscess as the main criterion, and later genomic criteria were used. However, although the strains caused abscesses, genetically they do not meet the criteria to be considered hypervirulent. Likewise, no experimental characterization was performed to evaluate their pathogenicity.

Reply: Thank you for your valuable feedback. We agree that experimental characterization is needed to confirm whether the strains are or are not hypervirulent. Unfortunately, we are not able to do *in vivo* or *in vitro* experiments and this was beyond the scope of this manuscript. Therefore we have not drawn hard conclusions on hvKp.

We have now removed the term ‘hvKp’ as much as possible for the isolates from this study throughout the manuscript and added the term ‘*putative*’ to the title.

We have now added this sentence to the introduction: *‘It has been suggested that the murine infection model is the gold-standard experimental approach for distinguishing hypervirulent from classical strains.(6, 11, 13, 14) However, this is rarely possible in routine diagnostic/surveillance settings’*

We added this to the discussion: *‘other studies often based the hvKp definition on the presence of a combination of certain virulence factors, a Kleborate virulence score of 5, detection of ST23, murine infection models and/or capsule type K1 or K2.’* and *‘Murine infection models for identifying hvKp are rarely possible in diagnostic/surveillance settings.’*

And this sentence to the limitation section in the discussion: *‘Finally, we did not perform in vitro or vivo studies to confirm hypervirulence or analyze associations of certain virulence factors with hypervirulence.’*

We have also added the limitation to the discussion that *in vivo* or *in vitro* studies are needed for conclusions on hypervirulence associations: *‘However, this study lacked a comparator non-hvKp group, did not include in vivo or in vitro experiments using isolates with and without these genes, and did not apply a clear definition of hvKp. Therefore, no firm conclusions can be drawn.’*

The study is relevant from the perspective of collecting clinical isolates causing various abscesses from patients hospitalized in various hospitals across the country. However, I believe that the question or objective of the study should be reconsidered. This is

because the fact that a strain causes an abscess does not necessarily mean that it must be hypervirulent. Perhaps the strains that caused metastasis in patients should be evaluated; these would have a greater chance of being hypervirulent, for that reason alone. It is understood that the definition of hypervirulence has changed over time; however, there are basic aspects that a strain must meet to be considered hypervirulent, ranging from having hypervirulence genes (contained in a plasmid) to having caused high mortality in experimental trials in laboratory mice. Likewise, the study should include analysis of variants that have been described in the literature in recent years; for example, convergent strains.

Reply: We agree that if a strain causes an abscess it does not necessarily mean that it must be hypervirulent. For this reason we had excluded isolates from non-spontaneous abscesses, such as related to recent surgery, but we agree that other patient factors may play a role in the development of an abscess. In the discussion, we have drawn no hard conclusions on hvKp and mentioned that the clinical picture depends on a combination of the virulence of the KpSC and patient characteristics.

We have now changed the objective of the study in the abstract, importance and introduction sections, e.g.: *'The main objective of this study was to determine occurrence and clinical, epidemiological and genomic characteristics of KpSC infections causing spontaneous abscesses in Dutch patients.'*

We have also changed the conclusions in the abstract, importance section and at the start and end of the discussion to match the objective, e.g.: *'When existing microbiological/molecular definitions would be used, several spontaneous abscesses could not be explained.'* and *'In total, 33 KpSC isolates from 33 patients with spontaneous abscesses were detected'* and *'When Kleborate virulence score, the proposed definition of Russo et al., (11) ST23 or capsular serotype were used to define hvKp molecularly/microbiologically, many spontaneous abscesses could not be explained.'*

We have now added this to the discussion: *'Non-hvKp may also cause abscesses in patients with certain underlying diseases'*

Furthermore, we have now removed the term 'hvKp' as much as possible for the isolates from this study throughout the manuscript and added the term '*putative*' to the title.

We only had five patients for which it was confirmed that there was more than one abscess location. We agree that for these isolates it is more likely that these are hvKp than for isolates that cause only one abscess. We have now added a section on these isolates to section 'Comparative genomic analysis for genes encoding putative virulence factors' in the results:

'Since it is more likely that the five patients with multiple abscess locations had true hvKp infections, we analyzed the virulence factors in these isolates separately. These isolates contained a median of 151 (IQR 107-161) virulence genes. Virulence genes that all these isolates contained were genes that were present in virtually all study isolates. Four of the five isolates contained salmochelin, 2/5 colibactin, 2/5 aerobactin (all had iutA) and 4/5 yersiniabactin gene clusters and 4/5 had rmpADC and 0/5 rmpA2. Kleborate scores were 0,1,2,4 and 5 and none of the isolates met the criteria of Russo et al.'

And this to the discussion:

'The Kleborate virulence score varied among patients with multiple abscess locations, that are more likely to have true hvKp infections (score of 0, 1, 2, 4, 5).'

'Among patients with multiple abscess locations, none met the definition of Russo et al..'

We agree that detection of hypervirulence genes and observed high mortality in experimental trials in laboratory mice are both needed to confirm hypervirulence (see answer to previous question above). However, this was beyond the scope of this manuscript.

The manuscript already includes analysis of recent variants, like convergent strains, but we agree that this can be expanded/emphasized:

- We already described plasmids from our isolates with both virulence factors and resistance genes (in section 'Plasmid analyses' of the results):
'Three plasmids from three isolates contained resistance genes, encoding genes associated with resistance against tetracyclines, aminoglycosides, trimethoprim and/or sulfonamides. Two of these three also contained virulence genes, encoding aerobactin, salmochelin, rmpADC, rmpA2, peg-344/pagO, traT, several less known genes identified as potential hvKp markers by Spadar et al.(40) and/or silver resistance.'
We already compared our plasmids to all plasmids from international *K. pneumoniae* species complex isolates with available sequence data via NCBI BLAST, which also includes plasmids from convergent strains.
- We did not find the clinically most relevant convergent strains having ESBL or carbapenemase genes as was already mentioned (in section 'Hypermucoviscosity and antimicrobial resistance' of the results), but we have added text to make this more clear: *'The isolates were susceptible to almost all antibiotics for which information was available (supplementary table 4). Among tested isolates, only two isolates had resistance to ciprofloxacin and one isolate had resistance to trimethoprim/sulfamethoxazole. Carbapenem-resistance, carbapenemase production or carbapenemase genes (*bla*_{NDM}, *bla*_{KPC}, *bla*_{VIM}, *bla*_{IMP}, or *bla*_{OXA-48}) were not found the isolates (supplementary table 4 and 5). We also found no extended-spectrum beta-lactamase (ESBL) genes, so we did not find the most clinically relevant convergent strains.'*
- Furthermore, we did compare our isolates with international hvKp or ST23 *K. pneumoniae* isolates (see M&M and figure 2), which also include several convergent strains like carbapenemase-producing hvKp, and we did not find a genetic cluster. We have now added this to emphasize this in the section 'Genetic relatedness' of the results:
*'The Dutch study isolates were compared to international *K. pneumoniae* (n=701), *K. variicola* (n=10) and *K. quasipneumoniae* (n=9) isolates that were hypervirulent and/or ST23 from the NCBI database, also including convergent strains, by wgMLST'*
- We have now added that this also included non-ST23 isolates in the section 'Genomic analysis' of M&M:
'Furthermore, the genomes of the isolates were compared to genomes of hypervirulent (also including other STs than ST23) and/or ST23 KpSC isolates from the National Center for Biotechnology Information (NCBI) database'
- We have now also added the ST23 hvKp isolates (including carbapenemase-producing hvKp) from the Irish outbreak, since these sequence data were not available yet at the time we searched for sequences. (Genomic and phylogenetic analysis of hypervirulent *Klebsiella pneumoniae* ST23 in Ireland | Microbiology Society).

Furthermore, we have added Illumina sequence data from some recent studies (including carbapenemase-producing hvKp):

Genomically defined hypervirulent Klebsiella pneumoniae contributed to early-onset increased mortality | Nature Communications

Deciphering the relative importance of genetic elements in hypervirulent Klebsiella pneumoniae to guide countermeasure development - ScienceDirect

Differentiation of hypervirulent and classical Klebsiella pneumoniae with acquired drug resistance | mBio

- We have now added this text to the introduction:
‘Unfortunately, the number of convergent hvKp infections seems to be increasing, but again different definitions are being used.(3, 16) Convergent hypervirulent Klebsiella pneumoniae strains are strains that combine both hypervirulence and multidrug resistance — two characteristics that historically existed in separate lineages.’
And this to section ‘Hypermucoviscosity and antimicrobial resistance’ of Results:
‘so we did not find the most clinically relevant convergent strains.’
And this to the conclusion in the discussion:
‘The most clinically relevant convergent hvKp strains, including CPE and ESBL, were not found.’

General Questions:

The introduction is described in reverse, i.e., the general questions at the end.

Reply: In our opinion, the general question should be put at the end, because the text above gives a general introduction on the topic and missing/contradicting information, which then result in the objective of the study.

Line 79: must be Klebsiella pneumoniae species complex (KpSC). Review Wyres, K. L., Lam, M. M. C. & Holt, K. E. Population genomics of Klebsiella pneumoniae. Nat. Rev. Microbiol. 18, 344-359 (2020).

Reply: Thank you for pointing this out, we have changed *Klebsiella pneumoniae* complex into *Klebsiella pneumoniae* species complex (KpSC) throughout the manuscript.

Line 144. The inclusion criteria for strains should be specified in the M&M.

Reply: We added the in- and exclusion criteria to the section ‘Bacterial isolates’ of M&M: *‘Meeting the clinical criteria was the inclusion criterium. Exclusion criteria were duplicates of the same person, non-KpSC species at confirmation and an unclear or not-confirmed presence of an abscess.’*

Line 125: The meaning of RIVM is unclear.

Reply: We replaced RIVM throughout the manuscript by *‘National Institute for Public Health and the Environment’*.

Lines 153-154. It is unclear which tests it refers to.

Reply: We tried to clarify this in the ‘Antimicrobial susceptibility testing’ section of M&M of the manuscript: *‘Furthermore, phenotypical susceptibility testing results (susceptible/susceptible with increased exposure/resistant) for nine selected antibiotics (see*

table 4), generated by the submitting laboratories in routine diagnostic procedures using EUCAST recommendations and cut-offs, were collected via the web-based questionnaire.'

Lines 160-175. It is important to identify the different virulence factors. Several platforms and assays are mentioned; however, Kleborate is designed for the identification of virulence factors related to Hv strains. What is the difference between the other systems, or what does the inclusion of canonical virulence factors contribute to the analysis if they are not ultimately discussed or contributed to the results or analysis? Kleborate 0-5 should describe their nature.

Reply: we noticed that in other articles on hvKp several different virulence factors and databases with virulence factors were analyzed. We were interested in all potential virulence factors that are described for *K. pneumoniae*, because it is not completely clear in our opinion which virulence factors contribute to hypervirulence. We examined all results to see whether we saw noteworthy associations. In our opinion, even if we did not find it relevant to discuss certain genes in the manuscript text, it could still be relevant to other researchers to add the results (it is part of the supplementary information, not of the main text and tables).

We don't know why certain virulence genes are added to the BIGSdb and not to the virulence factor database or the other way around, but our aim was to analyze all virulence factors in *K. pneumoniae*. Since these databases did not include the same virulence factors, we added both.

The nature of Kleborate score 0-5 is now further clarified in the section 'genomic analyses' in M&M: *'The Kleborate virulence score describes presence of key hypervirulence loci and ranges from 0 to 5: 0 = no yersiniabactin, colibactin or aerobactin; 1 = yersiniabactin only; 2 = yersiniabactin and colibactin (or colibactin only); 3 = aerobactin without yersiniabactin or colibactin; 4 = aerobactin with yersiniabactin (no colibactin); 5 = yersiniabactin, colibactin and aerobactin.'*

Line 189: Table S2, why should the rmpACD operon genes be separated?

Reply: we understand that these are part of the same operon, but we chose to mention the separate genes for all gene clusters and operons. We have now placed *rmpA*, *rmpC* and *rmpD* next to each other in table S1 and table S2 to make more clear these belong together.

Results:

Line 222. The exclusion criteria are unclear.

Reply: We added the in- and exclusion criteria to the section 'Bacterial isolates' of M&M: *'Meeting the clinical criteria was the inclusion criterium. Exclusion criteria were duplicates of the same person, non-KpSC species at confirmation and an unclear or not-confirmed presence of an abscess.'*

Likewise, the work actually identifies 12 potential HVs overall. The rest lack the genes described in Hv strains, and Kleborate confirms this with its virulence score of 0.1.2.

Reply: I am not sure which criteria you mean in defining those 12 hvKp, I assume you mean the criteria proposed by Russo *et al.*, which we have mentioned in the result section. There are differences between the definition proposed by Russo *et al.* and the Kleborate score. Therefore, in the results and discussion, we mention the results of different available

molecular hvKp criteria. If you think a molecular definition is missing, we would be happy to add that to the manuscript.

We have now removed the term 'hvKp' as much as possible for the isolates from this study throughout the manuscript and added the term '*putative*' to the title.

We have emphasized that the Kleborate score and the definition proposed by Russo *et al.* are important molecular definitions of hvKp, by adding the underlined words to the section 'genomic analyses' in the M&M:

'Two important molecular definitions of hvKp are Kleborate virulence score (11) and hvKp criteria recently proposed by Russo et al.(11) The Kleborate virulence score describes the presence of key hypervirulence loci' and 'Isolates were also checked for the hvKp criteria from Russo et al.'

and by adding the underlined words to the section 'Comparative genomic analysis for genes encoding putative virulence factors' of the results (and by starting this section with both of these definitions):

'Among the 33 KpSC isolates, 12 (36%) had the hypervirulence-associated maximum Kleborate virulence score of 5.' and 'Only twelve of 33 (36%) KpSC infections met the molecular hvKp definition recently proposed by Russo et al. where isolates should contain *iroB*, *iucA*, *peg-344*, *rmpA* and *rmpA2*.(11) These isolates had virulence scores ranging from 3 to 5. Seven isolates had both the maximum Kleborate score and met the definition of Russo et al. Ten isolates had no maximum Kleborate score and did not meet the definition of Russo et al. Differences between these two hvKp definitions were mainly due to differences in the *rmpA2* (n=5) gene and the colibactin (n=5) gene cluster.'

And by adding this to the discussion:

'Thirty-six percent had the hypervirulence-associated maximum Kleborate virulence score of 5 and 36% met the hvKp definition of Russo et al (*iroB*, *iucA*, *peg-344*, *rmpA* and *rmpA2*).(11) Seven isolates met both of these hvKp criteria.'

'Importantly, it is doubtful whether isolates with a low Kleborate virulence score and that do not meet the definition proposed by Russo et al. are true hvKp'

'When Kleborate virulence score, the proposed definition of Russo et al.,(10) ST23 or capsular serotype were used to define hvKp molecularly/microbiologically, many spontaneous abscesses could not be explained'

MALDI identifies 29 Kp and 4 Kv, and they mention that NGS identifies 28 Kpn, 4 Kv, and 1 Kq. Specifically, what analysis was used with the genome to identify the bacterial species? ANI?

Reply: We used the Kleborate result. We have now added that to the manuscript in the 'Antimicrobial susceptibility testing' and 'Genomic analyses' section of M&M and 'Occurrence' section of the results.

Figure 1 does not contribute much to the analysis. According to the study, the relationship between the presumed 12 Hv (score 5) and the others considered Hv could be seen. Based on the data presented, there are doubts about whether they can be considered Hv.

Reply: We respectfully disagree with the statement that figure 1 does not contribute to the analysis. In our opinion, this is a valuable figure, most importantly because *K. pneumoniae* species complex isolates from patients with spontaneous abscesses do not form a genetic cluster, so there seems to be no outbreak in the Netherlands.

We have now added an asterisk to visualize the isolates with Kleborate score 5 in figure 1.

We added this to the discussion: 'Importantly, it is doubtful whether isolates with a low Kleborate virulence score and that do not meet the definition proposed by Russo et al. are true hvKp'

Likewise, the use of MLST for Kpn for Kv is not reliable. There is evidence that does not contribute to your analysis (see doi:

<https://www.biorxiv.org/content/10.1101/2024.10.28.617863v1> or

<https://www.biorxiv.org/content/10.1101/2024.10.28.617863v1>). Likewise, the Kv MLST system should be used for the assignment of Kv STs. Likewise, include genomes of Hv strains described in other regions of the world for both Kq and Kv for comparison.

Reply: As described in the M&M, for MLST, we use schemes from SeqSphere. These include an MLST scheme from Institut Pasteur that is designed for all species within *K. pneumoniae* species complex, including *K. variicola*: <https://bigsd.biorpasteur.fr/klebsiella/>. We use this scheme and this is an internationally accepted scheme. SeqSphere does not include a scheme that is specifically developed for *K. variicola* and we found only 4 *K. variicola* isolates. It is beyond the scope of this study to develop a new scheme specifically for *K. variicola*.

For wgMLST, we also did not use a *K. variicola* specific-scheme, we used our in-house *K. pneumoniae* scheme. We have now added wgMLST data of *K. variicola* and *K. quasipneumoniae* isolates from NCBI and from the CPE surveillance to the manuscript. All species from *K. pneumoniae* species complex are checked for the percentage of detected core genes of the wgMLST scheme as quality control. WgMLST for species from *K. pneumoniae* species complex is only performed for isolates that have >90% of the core genes of the scheme. All *K. variicola* and *K. quasipneumoniae* isolates from this pilot (n=4 and n=1, resp.) and from our carbapenemase-producing surveillance (n=32 and n=29, resp.) and the international *K. variicola* isolates (n=10 and n=9 resp.) have >90% (*K. variicola* even >96%) of the core genes from our wgMLST scheme. Two *K. quasipneumoniae* isolates from our surveillance (*K. pneumoniae* in MALDI-TOF) were excluded from the wgMLST analysis due to a percentage detected core genes of 89.2% and 89.9%.

For the plasmid comparison with international plasmids, we also used NCBI BLAST en already included all KpSC species in our search.

We have now added separate minimum spanning trees for *K. pneumoniae*, *K. variicola* and *K. quasipneumoniae* using our in-house wgMLST scheme for KpSC (Figure 1 and 2 and Supplementary figures 1 to 5).

To clarify this, we added this to the 'Supplementary material and results' file: For classical multi-locus sequence typing (MLST), the existing KpSC schemes available via SeqSphere were used. For whole-genome wgMLST, minimum spanning trees were contemplated using BioNumerics version 8.1 using an in-house K. pneumoniae wgMLST scheme.(14) K. variicola and K. quasipneumoniae isolates were only included in the wgMLST analysis if they had >90% detected core genes of K. pneumoniae of the wgMLST scheme (two K. quasipneumoniae isolates from the CPE surveillance were excluded from the analysis due to a percentage of detected core genes of 89.2 % and 89.9%).

When a ST is new, you should request the assignment of the new ST from the curators of the corresponding database.

Reply: we have now requested the assignment of the new ST and changed ‘ new ST’ into ‘ST8481’ throughout the manuscript, figures and supplementary tables.

Lines 303-306: Is this an analysis of classical Hv vs. strains? What is the purpose of the comparison? This is unclear.

Reply:

This is the only large collection of sequenced *K. pneumoniae* species complex isolates from the Netherlands that we have at our institute. This included 1,762 sequenced CPE isolates (1,701 *K. pneumoniae*, 32 *K. variicola* and 29 *K. quasipneumoniae*) from the period February 2012 until October 2025. For the national carbapenemase-producing Enterobacterales surveillance, we also assess Kleborate virulence score. The large majority will probably be classical *K. pneumoniae*, only 4 CPE isolates (all *K. pneumoniae*) had Kleborate virulence score 5. We added these isolates to see whether there are genetic clusters of hvKp strains with these classical (or maybe hvKp) strains. We wondered whether there were convergent hvKp strains amongst the sequenced CPE isolates. We used wgMLST to analyze clusters.

We tried to explain this in the ‘genomic analyses’ section of the M&M: ‘Furthermore, the isolates were compared to KpSC isolates from the carbapenemase-producing Enterobacterales (CPE) surveillance of the National Institute for Public Health and the Environment(31) to examine genetic relatedness with this large sequenced collection of (mostly classical) KpSC isolates from the Netherlands. These 1,762 sequenced CPE isolates from the period February 2012 until October 2025 included only four *K. pneumoniae* isolates with maximum Kleborate virulence score 5.’

Line 308: Where are the Kq and Kv genomes? Likewise, these species should be compared separately with their HV genomes described in other studies.

Reply: See previous answer

Line 320. In your results, you describe what I mentioned earlier: identifying 12 strains as presumptive Hv strains. You describe 33 strains, as clearly showing that only these 12 met the criteria for presumptive Hv strains. Likewise, these strains should at least be characterized for their pathogenicity in a murine model, phagocytosis by macrophages, and resistance to human serum. Only then can they be confirmed as HV strains.

Reply: see previous answers on the definition of Russo *et al.*, Kleborate and experimental characterization.

I consider a hypervirulent *K. variicola* strain with a score of zero to be highly unlikely. In the case of *K. quasipneumoniae* strains that are described as HV, they have acquired both the virulence factors and the capsular serotype in their chromosome.

Reply: We agree it is unlikely that a *K. variicola* is hypervirulent when it has a Kleborate virulence score of 0. This isolate did not have any of the genes of the definition of Russo *et al* (*iroB*, *iucA*, *peg-344*, *rmpA* and *rmpA2*). It has no aerobactin, salmochelin, colibactin or yersiniabactin gene clusters, or *rmpADC* or *rmpA2*, but 99 other putative virulence genes. We

have now removed the term 'hvKp' as much as possible for the isolates from this study throughout the manuscript and added the term 'putative' to the title.

The *K. quasipneumoniae* isolate also has virulence score 0 and has several virulence genes, but not aerobactin, salmochelin, yersiniabactin, colibactin gene clusters, or *rmpA* or *rmpA2*. It did not have any of the genes of the definition of Russo et al (*iroB*, *iucA*, *peg-344*, *rmpA* and *rmpA2*). We found 4 plasmids in the *K. quasipneumoniae* isolate, among which one had putative virulence factors (only tellurite and silver resistance genes and a hypothetical protein).

Is there a relationship between string+ strains with the complete rmpACD operon? It should be noted that there is a third phenotype that few mention: rmpACD-negative strains that are positive to the string test. Do these strains exist in this study?

Reply: I had previously added this to the manuscript, but I removed it because the manuscript was already so long and it is already known that there is a relation between string test results and the rmpACD operon.

Among 19 isolates with a positive string test, 18 (95%) had a complete *rmpADC* operon, compared to 3 (21%) among 14 isolates with a negative string test (two isolates with truncated *rmpADC* locus were removed). There was one isolate with a positive string test that had no complete *rmpADC* operon (*rmpC* and *RmpD* genes were present, *rmpA2* was also not present).

I have now added this sentence to the section 'Comparative genomic analysis for genes encoding putative virulence factors' of the results: 'There was one isolate with a positive string test that had no complete rmpADC or rmpA2 locus (rmpC and RmpD genes were present).'

Discussion.

Various criteria have been described to effectively define an Hv strain; however, there is a strong consensus that the "murine infection model is the gold-standard experimental approach for distinguishing hypervirulent from classical strains."

Reply: see previous answer on experimental characterization.

Likewise, the use of the string test was effectively ruled out.

Reply: I had already mentioned this in the introduction: *'However hypermucoviscosity has no optimal sensitivity and specificity for detecting hvKp and a string test is therefore not reliable in defining hvKp.(7, 8).'*

I have now also added this sentence to the discussion and removed the string test from the rest of the discussion: 'A positive string test was also previously used, but there is strong evidence that this is not reliable in defining hvKp.(7, 8).'

The use of clinical data is now an interesting proposal, but it lacks clear criteria, not because of the data itself; rather, because the evolution of Hv KpSC is currently

underway, and significant discrepancies between virulence factors and pathogenicity in vivo have been reported.

Reply: We agree that hv KpSC are changing. Virulence genes causing hypervirulence can also change. In addition, insights into virulence factors and definitions will change. Clinical data and molecular data will probably never completely match. In our opinion, this study adds valuable information to the available literature. Unfortunately, we are not able to do *in vivo* or *in vitro* experiments, but we have now added this limitation (as described above).

Likewise, the convergence phenomenon is an important factor to consider, adding variability, which would be the low pathogenicity in MDR Hv-KpSc strains.

Reply: We did not find MDR Hv-KpSC strains (no carbapenemase or ESBL genes were found). We acknowledge that antimicrobial resistance could affect the virulence of a hvKp. However, we have only included KpSC isolates from patients with abscesses, so convergent hvKp strains in which the pathogenicity is so strongly affected that abscesses are no longer present are not included.

We have now added this to the discussion:

'Furthermore, convergent hvKp strains in which the pathogenicity is so strongly affected by addition of antimicrobial resistance genes that abscesses are no longer present are not included.'

For further discussion about convergence, see our previous answer.

Ultimately, the discussion should be adjusted according to the considerations of the review.

The data presented on presumably hypervirulent *K. pneumoniae* species complex in your country appear to be typical hypervirulent strains, it shouldn't be much of a problem to characterize them in greater depth.

Reply: see previous answers.

Re: Spectrum02259-25R2 (**Genomic epidemiology of putative hypervirulent *Klebsiella pneumoniae* species complex in Dutch patients, January-December 2022**)

Dear Dr. Karuna E.W. Vendrik:

Your manuscript has been accepted, and I am forwarding it to the ASM production staff for publication. Your paper will first be checked to make sure all elements meet the technical requirements. ASM staff will contact you if anything needs to be revised before copyediting and production can begin. Otherwise, you will be notified when your proofs are ready to be viewed.

Sincerely,
Gabriele Arcari
Editor
Microbiology Spectrum